# RE-IMAGEN: RETRIEVAL-AUGMENTED TEXT-TO-IMAGE GENERATOR

**Wenhu Chen, Hexiang Hu, Chitwan Saharia, William W. Cohen**
Google Research
{wenhuchen,hexiang,sahariac,wcohen}@google.com

## ABSTRACT

Research on text-to-image generation has witnessed significant progress in generating diverse and photo-realistic images, driven by diffusion and auto-regressive models trained on large-scale image-text data. Though state-of-the-art models can generate high-quality images of common entities, they often have difficulty generating images of uncommon entities, such as 'Chortai (dog)' or 'Picarones (food)'. To tackle this issue, we present the Retrieval-Augmented Text-to-Image Generator (Re-Imagen), a generative model that uses retrieved information to produce high-fidelity and faithful images, even for rare or unseen entities. Given a text prompt, Re-Imagen accesses an external multi-modal knowledge base to retrieve relevant (image, text) pairs and uses them as references to generate the image. With this retrieval step, Re-Imagen is augmented with the knowledge of high-level semantics and low-level visual details of the mentioned entities, and thus improves its accuracy in generating the entities' visual appearances. We train Re-Imagen on a constructed dataset containing (image, text, retrieval) triples to teach the model to ground on both text prompt and retrieval. Furthermore, we develop a new sampling strategy to interleave the classifier-free guidance for text and retrieval conditions to balance the text and retrieval alignment. Re-Imagen achieves significant gain on FID score over COCO and WikiImage. To further evaluate the capabilities of the model, we introduce EntityDrawBench, a new benchmark that evaluates image generation for diverse entities, from frequent to rare, across multiple object categories including dogs, foods, landmarks, birds, and characters. Human evaluation on EntityDrawBench shows that Re-Imagen can significantly improve the fidelity of generated images, especially on less frequent entities.

## 1 INTRODUCTION

Recent research efforts on conditional generative modeling, such as Imagen (Saharia et al., 2022), DALL·E 2 (Ramesh et al., 2022), and Parti (Yu et al., 2022), have advanced text-to-image generation to an unprecedented level, producing accurate, diverse, and even create images from text prompts. These models leverage paired image-text data at Web scale (with hundreds of millions of training examples), and powerful backbone generative models, *i.e.*, autoregressive models (Van Den Oord et al., 2017; Ramesh et al., 2021; Yu et al., 2022), diffusion models (Ho et al., 2020; Dhariwal & Nichol, 2021), etc, and generate highly realistic images. Studying these models' generation results, we discovered their outputs are surprisingly sensitive to the frequency of the entities (or objects) in the text prompts. In particular, when generating text prompts about frequent entities, these models often generate realistic images, with faithful grounding to the entities' visual appearance. However, when generating from text prompts with less frequent entities, those models either hallucinate non-existent entities, or output related frequent entities (see Figure 1), failing to establish a connection between the generated image and the visual appearance of the mentioned entity. This key limitation can greatly harm the trustworthiness of text-to-image models in real-world applications and even raise ethical concerns. In our studies, we found these models suffer from significant quality degradation in generating visual objects associated with under-represented groups.

In this paper, we propose a **Re**trieval-augmented Text-to-**Ima**ge **Gen**erator (Re-Imagen), which alleviates such limitations by searching for entity information in a multi-modal knowledge base, rather than attempting to memorize the appearance of rare entities. Specifically, we define our multi-modal

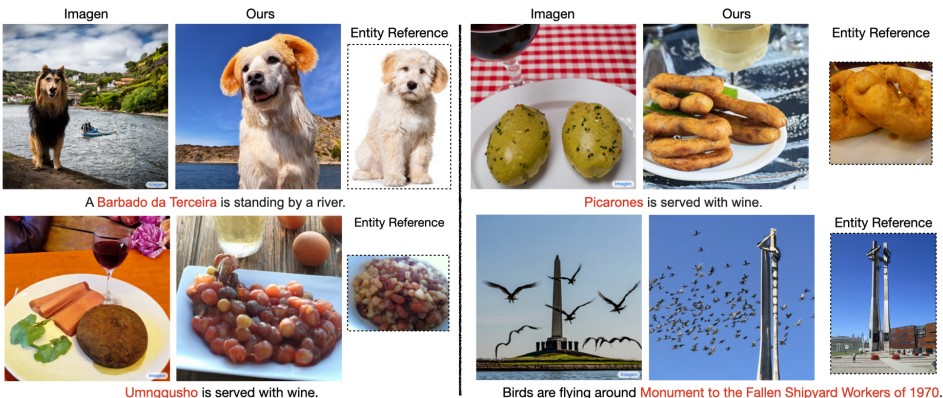

Figure 1: Comparison of images generated by Imagen and Re-Imagen on less frequent entities. We observe that Imagen hallucinates the entities while Re-Imagen maintains better faithfulness.

knowledge base encodes the visual appearances and descriptions of entities with a collection of reference <image, text> pairs'. To use this resource, Re-Imagen first uses the input text prompt to retrieve the most relevant <image, text> pairs from the external multi-modal knowledge base, then uses the retrieved knowledge as model additional inputs to synthesize the target images. Consequently, the retrieved references provide knowledge regarding the semantic attributes and the concrete visual appearance of mentioned entities to guide Re-Imagen to paint the entities in the target images.

The backbone of Re-Imagen is a cascaded diffusion model (Ho et al., 2022), which contains three independent generation stages (implemented as U-Nets (Ronneberger et al., 2015)) to gradually produce high-resolution (*i.e.*, 1024×1024) images. In particular, we train Re-Imagen on a dataset constructed from the image-text dataset used by Imagen (Saharia et al., 2022), where each data instance is associated with the top-k nearest neighbors within the dataset, based on text-only BM25 score. The retrieved top-k <image, text> pairs will be used as a reference for the model attend to. During inference, we design an interleaved guidance schedule that switches between text guidance and retrieval guidance, which ensures both text alignment and entity alignment. We show some examples generated by Re-Imagen, and compare them against Imagen in Figure 1. We can qualitatively observe that our images are more faithful to the appearance of the reference entity.

To further quantitatively evaluate Re-Imagen, we present *zero-shot text-to-image generation* results on two challenging datasets: COCO (Lin et al., 2014) and WikiImages (Chang et al., 2022)[1]. Re-Imagen uses an external non-overlapping image-text database as the knowledge base for retrieval and then grounds on the retrieval to synthesize the target image. We show that Re-Imagen achieves the state-of-the-art performance for text-to-image generation on COCO and WikiImages, measured in FID score (Heusel et al., 2017), among non-fine-tuned models. For the non-entity-centric dataset COCO, the performance gain is coming from biasing the model to generate images with similar styles as the retrieved in-domain images. For the entity-centric dataset WikiImages, the performance gain comes from grounding the generation on retrieved images containing similar entities. We further evaluate Re-Imagen on a more challenging benchmark — EntityDrawBench, to test the model's ability to generate a variety of infrequent entities (dogs, landmarks, foods, birds, animated characters) in different scenes. We compare Re-Imagen with Imagen (Saharia et al., 2022), DALL-E 2 (Ramesh et al., 2022) and StableDiffusion (Rombach et al., 2022) in terms of faithfulness and photorealism with human raters. We demonstrate that Re-Imagen can generate faithful and realistic images on 80% over input prompts, beating the existing best models by at least 30% on EntityDrawBench. Analysis shows that the improvements are mostly coming from low-frequency visual entities.

To summarize, our key contributions are: (1) a novel retrieval-augmented text-to-image model Re-Imagen, which improves FID scores on two datasets; (2) interleaved classifier-free guidance during sampling to ensure both text alignment and entity fidelity; and (3) We introduce EntityDrawBench and show that Re-Imagen can significantly improve faithfulness on less-frequent entities.

---

[1]The original WikiImages database contains (entity image, entity description) pairs. It was a crawled from Wikimedia Commons for visual question answering, and we repurpose it here for text-to-image generation.

## 2    RELATED WORK

**Text-to-Image Diffusion Models** There has been a wide-spread success (Ashual et al., 2022; Ramesh et al., 2022; Saharia et al., 2022; Nichol et al., 2021) in modeling text-to-image generation with diffusion models, which has outperformed GANs (Goodfellow et al., 2014) and auto-regressive Transformers (Ramesh et al., 2021) in photorealism and diversity (under similar model size), without training instability and mode collapsing issues. Among them, some recent large text-to-image models such as Imagen (Saharia et al., 2022), GLIDE (Nichol et al., 2021), and DALL-E2 (Ramesh et al., 2022) have demonstrated excellent generation from complex prompt inputs. These models achieve highly fine-grained control over the generated images with text inputs. However, they do not perform explicit grounding over external visual knowledge and are restricted to memorizing the visual appearance of every possible visual entity in their parameters. This makes it difficult for them to generalize to rare or even unseen entities. In contrast, Re-Imagen is designed to free the diffusion model from memorizing, as models are encouraged to retrieve semantic neighbors from the knowledge base and use retrievals as context to paint the image. Re-Imagen improves the grounding of the diffusion models to real-world knowledge and is therefore capable of faithful image synthesis.

**Concurrent Work** There are several concurrent works (Li et al., 2022; Blattmann et al., 2022; Ashual et al., 2022), that also leverage retrieval to improve diffusion models. RDM (Blattmann et al., 2022) is trained similarly to Re-Imagen, using examples and near neighbors, but the neighbors in RDM are selected using image features, and at inference time retrievals are replaced with user-chosen exemplars. RDM was shown to effectively transfer artistic style from exemplars to generated images. In contrast, our proposed Re-Imagen conditions on both text and multi-modal neighbors to generate the image includes retrieval at inference time and is demonstrated to improve performance on rare images (as well as more generally). KNN-Diffusion (Ashual et al., 2022) is more closely related work to us, as it also uses retrieval to the quality of generated images. However, KNN-Diffusion uses discrete image representations, while Re-Imagen uses the raw pixels, and Re-Imagen's retrieved neighbors can be <image, text> pairs, while KNN-Diffusion's are only images. Quantitatively, Re-Imagen outperforms KNN-Diffusion on the COCO dataset significantly.

**Others** Due to the space limit, we provide an additional literature review in Appendix A.

## 3    MODEL

In this section, we start with background knowledge, in the form of a brief overview of the cascaded diffusion models used by Imagen. Next, we describe the concrete technical details of how we incorporate retrieval for Re-Imagen. Finally, we discuss interleaved guidance sampling.

### 3.1    PRELIMINARIES

**Diffusion Models** Diffusion models (Sohl-Dickstein et al., 2015) are latent variable models, parameterized by $\theta$, in the form of $p_\theta(\boldsymbol{x}_0) := \int p_\theta(\boldsymbol{x}_{0:T}) d\boldsymbol{x}_{1:T}$, where $\boldsymbol{x}_1, \cdots, \boldsymbol{x}_T$ are "noised" latent versions of the input image $\boldsymbol{x}_0 \sim q(\boldsymbol{x}_0)$. Note that the dimensionality of both latents and the image is the same throughout the entire process, with $\boldsymbol{x}_{0:T} \in \mathbb{R}^d$ and $d$ equals the product of <height, width, # of channels>. The process that computes the posterior distribution $q(\boldsymbol{x}_{1:T}|\boldsymbol{x}_0)$ is also called the forward (or diffusion) process, and is implemented as a predefined Markov chain that gradually adds Gaussian noise to the data according to a schedule $\beta_t$:

$$q(\boldsymbol{x}_{1:T}|\boldsymbol{x}_0) = \prod_{t=1}^{T} q(\boldsymbol{x}_t|\boldsymbol{x}_{t-1}) \qquad q(\boldsymbol{x}_t|\boldsymbol{x}_{t-1}) := \mathcal{N}(\boldsymbol{x}_t; \sqrt{1-\beta_t}\boldsymbol{x}_{t-1}, \beta_t \boldsymbol{I}) \tag{1}$$

Diffusion models are trained to learn the image distribution by reversing the diffusion Markov chain. Theoretically, this reduces to learning to denoise $\boldsymbol{x}_t \sim q(\boldsymbol{x}_t|\boldsymbol{x}_0)$ into $\boldsymbol{x}_0$, with a time re-weighted square error loss—see Ho et al. (2020) for the complete proof:

$$\mathbb{E}_{\boldsymbol{x}_0, \boldsymbol{\epsilon}, t}[w_t \cdot ||\hat{\boldsymbol{x}}_\theta(\boldsymbol{x}_t, \boldsymbol{c}) - \boldsymbol{x}_0||_2^2] \tag{2}$$

Here, the noised image is denoted as $\boldsymbol{x}_t := \sqrt{\bar{\alpha}_t}\boldsymbol{x}_0 + \sqrt{1-\bar{\alpha}_t}\boldsymbol{\epsilon}$, $\boldsymbol{x}_0$ is the ground-truth image, $\boldsymbol{c}$ is the condition, $\boldsymbol{\epsilon} \sim \mathcal{N}(\boldsymbol{0}, \boldsymbol{I})$ is the noise term, $\alpha_t := 1 - \beta_t$ and $\bar{\alpha}_t := \prod_{s=1}^{t} \alpha_s$. To simplify notation,

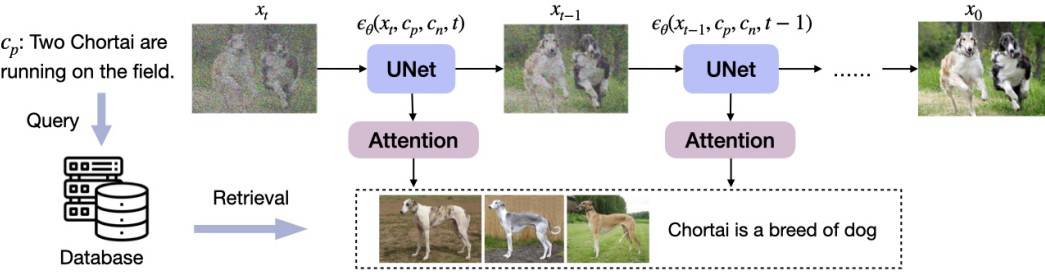

Figure 2: An illustration of the text-to-image generation pipeline in the $64\times$ diffusion model. Specifically, Re-Imagen learns a UNet to iteratively predict $\epsilon(\boldsymbol{x_t}, \boldsymbol{c_n}, \boldsymbol{c_p}, t)$ that denoises the image. ($\boldsymbol{c_n}$: a set of retrieved image-text pairs from the database; $\boldsymbol{c_p}$: input text prompt; $t$: current time-step)

we will allow the condition $\boldsymbol{c}$ to include multiple conditioning signals, such as text prompts $\boldsymbol{c_p}$, a low-resolution image input $\boldsymbol{c_x}$ (which is used in super-resolution), or retrieved neighboring images $\boldsymbol{c_n}$ (which are used in Re-Imagen). Imagen (Saharia et al., 2022) uses a U-Net (Ronneberger et al., 2015) to implement $\epsilon_\theta(\boldsymbol{x_t}, \boldsymbol{c}, t)$. The U-Net represents the reversed noise generator as follows:

$$\hat{\boldsymbol{x}}_\theta(\boldsymbol{x_t}, \boldsymbol{c}) := (\boldsymbol{x_t} - \sqrt{1 - \bar{\alpha}_t}\boldsymbol{\epsilon}_\theta(\boldsymbol{x_t}, \boldsymbol{c}, t))/\sqrt{\bar{\alpha}_t} \tag{3}$$

During the training, we randomly sample $t \sim \mathcal{U}([0, 1])$ and image $\boldsymbol{x}_0$ from the dataset $\mathcal{D}$, and minimize the difference between $\hat{\boldsymbol{x}}_\theta(\boldsymbol{x_t}, \boldsymbol{c})$ and $\boldsymbol{x}_0$ according to Equation 2. At the inference time, the diffusion model uses DDPM (Ho et al., 2020) to sample recursively as follows:

$$\boldsymbol{x}_{t-1} = \frac{\sqrt{\bar{\alpha}_{t-1}}\beta_t}{1 - \bar{\alpha}_t}\hat{\boldsymbol{x}}_\theta(\boldsymbol{x_t}, \boldsymbol{c}) + \frac{\sqrt{\alpha_t}(1 - \bar{\alpha}_{t-1})}{1 - \bar{\alpha}_t}\boldsymbol{x_t} + \frac{\sqrt{(1 - \bar{\alpha}_{t-1})\beta_t}}{\sqrt{1 - \bar{\alpha}_t}}\boldsymbol{\epsilon} \tag{4}$$

The model sets $\boldsymbol{x}_T$ as a Gaussian noise with $T$ denoting the total number of diffusion steps, and then keeps sampling in reverse until step $T = 0$, i.e. $\boldsymbol{x}_T \to \boldsymbol{x}_{T-1} \to \cdots$, to reach the final image $\hat{\boldsymbol{x}}_0$.

For better generation efficiency, cascaded diffusion models (Ho et al., 2022; Ramesh et al., 2022; Saharia et al., 2022) use three separate diffusion models to generate high-resolution images gradually, going from low resolution to high resolution. The three models $64\times$ model, $256\times$ super-resolution model, and $1024\times$ super-resolution model gradually increase the model resolution to $1024 \times 1024$.

**Classifier-free Guidance** Ho & Salimans (2021) first proposed classifier-free guidance to trade off diversity and sample quality. This sampling strategy has been widely used due to its simplicity. In particular, Imagen (Saharia et al., 2022) adopts an adjusted $\epsilon$-prediction as follows:

$$\hat{\boldsymbol{\epsilon}} = w \cdot \boldsymbol{\epsilon}_\theta(\boldsymbol{x_t}, \boldsymbol{c}, t) - (w - 1) \cdot \boldsymbol{\epsilon}_\theta(\boldsymbol{x_t}, t) \tag{5}$$

where $w$ is the guidance weight. The unconditional $\epsilon$-prediction $\boldsymbol{\epsilon}_\theta(\boldsymbol{x_t}, t)$ is calculated by dropping the condition, i.e. the text prompt.

### 3.2 Generating Image with Multi-Modal Knowledge

Similar to Imagen (Saharia et al., 2022), Re-Imagen is a cascaded diffusion model, consisting of $64\times$, $256\times$, and $1024\times$ diffusion models. However, Re-Imagen augments the diffusion model with the new capability of leveraging multimodal 'knowledge' from the external database, thus freeing the model from memorizing the appearance of rare entities. For brevity (and concreteness) we present below a high-level overview of the $64\times$ model: the others are similar.

**Main Idea** As shown in Figure 2, during the denoising process, Re-Imagen conditions its generation result not only on the text prompt $\boldsymbol{c_p}$ (and also with $\boldsymbol{c_x}$ for super-resolution), but on the neighbors $\boldsymbol{c_n}$ that were retrieved from the external knowledge base. Here, the text prompt $\boldsymbol{c_p} \in \mathbb{R}^{n \times d}$ is represented using a T5 embedding (Raffel et al., 2020), with $n$ being the text length and $d$ being the embedding dimension. Meanwhile, the top-k neighbors $\boldsymbol{c_n} := [<\text{image, text}>_1, \cdots, <\text{image, text}>_k]$ are retrieved from external knowledge base $\mathcal{B}$, using the input prompt $p$ as the query and a retrieval similarity function $\gamma(p, \mathcal{B})$. We experimented with two different choices for the similarity function: maximum inner product scores for BM25 (Robertson et al., 2009) and CLIP (Radford et al., 2021).

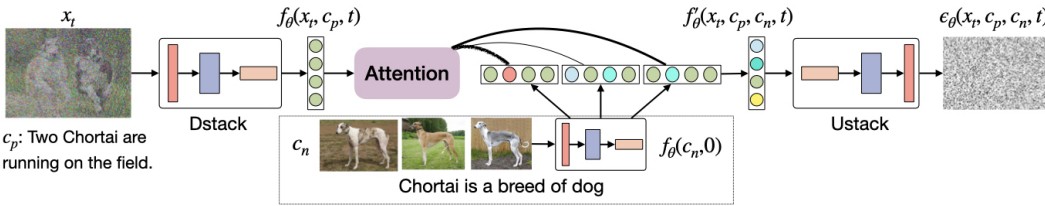

Figure 3: The detailed architecture of our model. The retrieved neighbors are first encoded using the DStack encoder and then used to augment the intermediate representation of the denoising image (via cross-attention). The augmented representation is fed to the UStack to predict the noise.

**Model Architecture** We show the architecture of our model in Figure 3, where we decompose the UNet into the downsampling encoder (DStack) and the upsampling decoder (UStack). Specifically, the DStack takes an image, a text, and a time step as the input, and generates a feature map, which is denoted as $f_\theta(\boldsymbol{x}_t, \boldsymbol{c}_p, t) \in \mathbb{R}^{F \times F \times d}$, with $F$ denoting the feature map width and $d$ denoting the hidden dimension. We share the same DStack encoder when we encode the retrieved <image, text> pairs (with $t$ set to zero) which produce a set of feature maps $f_\theta(\boldsymbol{c}_n, 0) \in \mathbb{R}^{K \times F \times F \times d}$. We then use a multi-head attention module (Vaswani et al., 2017) to extract the most relevant information to produce a new feature map $f'_\theta(\boldsymbol{x}_t, \boldsymbol{c}_p, \boldsymbol{c}_n, t) = Attn(f_\theta(\boldsymbol{x}_t, \boldsymbol{c}_p, t), f_\theta(\boldsymbol{c}_n, 0))$. The upsampling stack decoder then predicts the noise term $\boldsymbol{\epsilon}_\theta(\boldsymbol{x}_t, \boldsymbol{c}_p, \boldsymbol{c}_n, t)$ and uses it to compute $\hat{\boldsymbol{x}}_\theta$ with Equation 3, which is either used for regression during training or DDPM sampling.

**Model Training** In order to train Re-Imagen, we construct a new dataset KNN-ImageText based on the 50M ImageText-dataset used in Imagen. There are two motivations for selecting this dataset. (1) the dataset contains many similar photos regarding specific entities, which is extremely helpful for obtaining similar neighbors, and (2) the dataset is highly sanitized with fewer unethical or harmful images. For each instance in the 50M ImageText-dataset, we search over the same dataset with text-to-text BM25 similarity to find the top-2 neighbors as $\boldsymbol{c}_n$ (excluding the query instance). We experimented with both CLIP and BM25 similarity scores, and retrieval was implemented with ScaNN (Guo et al., 2020). We train Re-Imagen on the KNN-ImageText by minimizing the loss function of Equation 2. During training, we also randomly drop the text and neighbor conditions independently with 10% chance. Such random dropping will help the model learn the marginalized noise term $\boldsymbol{\epsilon}_\theta(\boldsymbol{x}_t, \boldsymbol{c}_p, t)$ and $\boldsymbol{\epsilon}_\theta(\boldsymbol{x}_t, \boldsymbol{c}_n, t)$, which will be used for the classifier-free guidance.

**Interleaved Classifier-free Guidance** Different from existing diffusion models, our model needs to deal with more than one condition, *i.e.*, text prompts $\boldsymbol{c}_t$ and retrieved neighbors $\boldsymbol{c}_n$, which allows new options for incorporating guidance. In particular, Re-Imagen could use classifier-free guidance by subtracting the unconditioned $\epsilon$-predictions, or either of the two partially conditioned $\epsilon$-predictions. Empirically, we observed that subtracting unconditioned $\epsilon$-predictions (the standard classifier-free guidance of Figure 3.1) often leads to an undesired imbalance, where the outputs are either dominated by the text condition or the neighbor condition. Hence, we designed an interleaved guidance schedule that balances the two conditions. Formally, we define the two adjusted $\epsilon$-predictions as:

$$\hat{\boldsymbol{\epsilon}}_p = w_p \cdot \boldsymbol{\epsilon}_\theta(\boldsymbol{x}_t, \boldsymbol{c}_p, \boldsymbol{c}_n, t) - (w_p - 1) \cdot \boldsymbol{\epsilon}_\theta(\boldsymbol{x}_t, \boldsymbol{c}_n, t)$$
$$\hat{\boldsymbol{\epsilon}}_n = w_n \cdot \boldsymbol{\epsilon}_\theta(\boldsymbol{x}_t, \boldsymbol{c}_p, \boldsymbol{c}_n, t) - (w_n - 1) \cdot \boldsymbol{\epsilon}_\theta(\boldsymbol{x}_t, \boldsymbol{c}_p, t)$$

(6)

where $\hat{\boldsymbol{\epsilon}}_p$ and $\hat{\boldsymbol{\epsilon}}_n$ are the text-enhanced and neighbor-enhanced $\epsilon$-predictions, respectfully. Here, $w_p$ is the text guidance weight and $w_n$ is the neighbor guidance weight. We then interleave the two guidance predictions by a certain predefined ratio $\eta$. Specifically, at each guidance step, we sample a $[0, 1]$-uniform random number $R$, and $R < \eta$, we use $\hat{\boldsymbol{\epsilon}}_p$, and otherwise $\hat{\boldsymbol{\epsilon}}_n$. We can adjust $\eta$ to balance the faithfulness w.r.t text description or the retrieved image-text pairs.

## 4 EXPERIMENTS

Re-Imagen consists of three submodels: a 2.5B 64×64 text-to-image model, a 750M 256×256 super-resolution model and a 400M 1024×1024 super-resolution model. We also have a Re-Imagen-small with 1.4B 64×64 text-to-image model to understand the impact of model size.

| Model | Params | COCO | | WikiImages | |
|---|---|---|---|---|---|
| | | FID | ZS FID | FID | ZS FID |
| GLIDE (Nichol et al., 2021) | 5B | - | 12.24 | - | - |
| DALL-E 2 (Ramesh et al., 2022) | ~5B | - | 10.39 | - | - |
| Stable-Diffusion (Rombach et al., 2022) | 1B | - | 12.63 | - | 7.50 |
| Imagen (Saharia et al., 2022) | 3B | - | 7.27 | - | 6.44 |
| Make-A-Scene (Gafni et al., 2022) | 4B | 7.55$^\dagger$ | 11.84 | - | - |
| Parti (Yu et al., 2022) | 20B | **3.22**$^\dagger$ | 7.23 | - | - |
| *Retrieval-Augmented Model* | | | | | |
| KNN-Diffusion (Ashual et al., 2022) | - | 16.66 | - | - | - |
| Memory-Driven Text-to-Image (Li et al., 2022) | - | 19.47 | - | - | - |
| Re-Imagen ($\gamma$=BM25; $\mathcal{B}$=IND; $k$=2) | 3.6B | **5.25** | - | **5.88** | - |
| Re-Imagen ($\gamma$=BM25; $\mathcal{B}$=OOD; $k$=2) | 3.6B | - | **6.88** | - | **5.80** |
| Re-Imagen-small ($\gamma$=BM25; $\mathcal{B}$=IND; $k$=2) | 2.4B | 5.73 | - | 6.32 | - |
| Re-Imagen-small ($\gamma$=BM25; $\mathcal{B}$=OOD; $k$=2) | 2.4B | - | 7.32 | - | 6.04 |

Table 1: MS-COCO results for text-to-image generation. We use a guidance weight of 1.25 for the $64\times$ diffusion model and 5 for our $256\times$ super-resolution model. († *is fine-tuned*)

We finetune these models on the constructed KNN-ImageText dataset. We evaluate the model under two settings: (1) automatic evaluation on COCO and WikiImages dataset, to measure the model's general performance to generate photorealistic images, and (2) human evaluation on the newly introduced EntityDrawBench, to measure the model's capability to generate long-tail entities.

**Training and Evaluation details** The fine-tuning was run for 200K steps on 64 TPU-v4 chips and completed within two days. We use Adafactor for the $64\times$ model and Adam for the $256\times$ super-resolution model with a learning rate of 1e-4. We set the number of neighbors $k$=2 and set $\gamma$=BM25 during training. For the image-text database $\mathcal{B}$, we consider three different variants: (1) the in-domain training set, which contains non-overlapping small-scale in-domain image-text pairs from COCO or WikiImages, (2) the out-of-domain LAION dataset (Schuhmann et al., 2021) containing 400M <image, text> crawled pairs. Since indexing ImageText and LAION with CLIP encodings is expensive, we only considered the BM25 retriever for these databases.

## 4.1 EVALUATION ON COCO AND WIKIIMAGES

In these two experiments, we used the standard non-interleaved classifier-free guidance (Figure 3.1) with $T$=1000 steps for both the $64\times$ diffusion model and $256\times$ super-resolution model. The guidance weight $w$ for the $64\times$ model is swept over [1.0, 1.25, 1.5, 1.75, 2.0], while the $256\times256$ super-resolution models' guidance weight $w$ is swept over [1.0, 5.0, 8.0, 10.0]. We select the guidance $w$ with the best FID score, which is reported in Table 1. We also demonstrate examples in Figure 4.

**COCO Results** COCO is the most widely-used benchmark for text-to-image generation models. Although COCO does not contain many rare entities, it does contain unusual combinations of common entities, so it is plausible that retrieval augmentation could also help with some challenging text prompts. We adopt FID (Heusel et al., 2017) score to measure image quality. Following the previous literature, we randomly sample 30K prompts from the validation set as input to the model. The generated images are compared with the reference images from the full validation set (42K). We list the results in two columns: FID-30K denotes that the model with access to the in-domain COCO train set, while Zero-shot FID-30K does not have access to any COCO data.

Re-Imagen can achieve a significant gain on FID by retrieving from external databases: roughly a 2.0 absolute FID improvement over Imagen. Its performance is even better than fine-tuned Make-A-Scene (Gafni et al., 2022). We found that Re-Imagen retrieving from OOD database achieves less gain than IND database, but still obtains a 0.4 FID improvement over Imagen. When comparing with other retrieval-augmented models, Re-Imagen is shown to outperform KNN-Diffusion and Memory-Driven T2I models by a significant margin of 11 FID score. We also note that Re-Imagen-small is also competent in the FID, which outperforms normal-sized Imagen with fewer parameters.

As COCO does not contain infrequent entities, retrievals from the in-domain database mainly provide useful 'style knowledge' for the model to ground on. Re-Imagen can better adapt to COCO

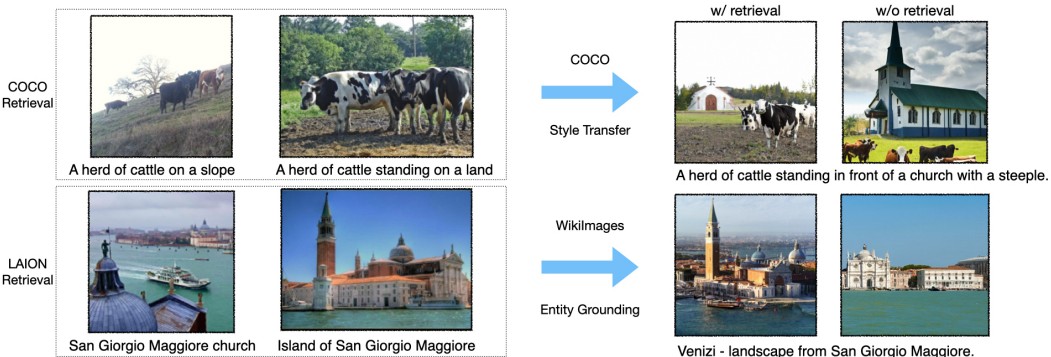

Figure 4: The retrieved top-2 neighbors of COCO and WikiImages and model generation.

distribution, thus achieving a better FID score. As can be seen in the upper part of from Figure 4, Re-Imagen with retrieval generates images of the same style as COCO, while without retrieval, the output is still high quality, but the style is less similar to COCO.

**WikiImages Results** WikiImages is constructed based on the multimodal corpus provided in Web-QA (Chang et al., 2022), which consists of <image, text> pairs crawled from Wikimedia Commons[2]. We filtered the original corpus to remove noisy data (see AppendixB), which leads to a total of 320K examples. We randomly sample 22K as our validation set to perform zero-shot evaluation, we further sample 20K prompts from the dataset as the input. Similar to the previous experiment, we also adopt the guidance weight schedule as before and evaluate 256×256 images.

From Table 1, we found that using the OOD database (LAION) actually achieves better performance than using the IND database. Unlike COCO, WikiImages contains mostly entity-focused images, thus the importance of finding relevant entities in the database is more important than distilling the styles from the training set—and since the scale of LAION-400M is 100x larger than an in-domain database, the chance of retrieving related entities is much higher, which leads to better performance. One example is depicted in the lower part of Figure 4, where the LAION retrieval finds 'Island of San Giorgio Maggiore', which helps the model generate the classical Renaissance-style church.

## 4.2 ENTITY FOCUSED EVALUATION ON ENTITYDRAWBENCH

**Dataset Construction** We introduce EntityDrawBench to evaluate the model's capability to generate diverse sets of entities in different visual scenes. Specifically, we pick various types of visual entities (dog breeds, landmarks, foods, birds, and animated characters) from Wikipedia Commons, Google Landmarks and Fandom to construct our prompts. In total, we collect 250 entity-centric prompts for evaluation. These prompts are mostly very unique and we cannot find them on the Internet, let alone the model training data. The dataset construction details are in Appendix C. To evaluate the model's capability to ground on broader types of entities, we also randomly select 20 objects like 'sunglasses, backpack, vase, teapot, etc' and write creative prompts for them. We compare our generation results with the results from DreamBooth (Ruiz et al., 2022) in Appendix H.

We use the constructed prompt as the input and its corresponding image-text pairs as the 'retrieval' for Re-Imagen, to generate four 1024×1024 images. For the other models, we feed the prompts directly also to generate four images. We pick the best image of 4 random samples to rate its Photorealism and Faithfulness by human raters. For photorealism, we rate 1 if the image is moderately realistic without noticeable artifacts. For the faithfulness measure, we rate 1 if the image is faithful to both the entity appearance and the text description.

**EntityDrawBench Results** We use the proposed interleaved classifier-free guidance (subsection 3.2) for the 64× diffusion model, which runs for 256 diffusion steps under a strong guidance weight of $w$=30 for both text and neighbor conditions. For the 256× and 1024× resolution models, we use a constant guidance weight of 5.0 and 3.0, respectively, with 128 and 32 diffusion steps.

---

[2]https://commons.wikimedia.org/wiki/Main_Page

| Model | Faithfulness | | | | | | Photorealism |
|---|---|---|---|---|---|---|---|
| | Dogs | Foods | Landmarks | Birds | Characters | Broader | All |
| Imagen | 0.28 | 0.26 | 0.27 | 0.84 | 0.10 | 0.54 | 0.98 |
| DALL-E 2 | 0.60 | 0.47 | 0.36 | 0.82 | 0.08 | 0.58 | 0.98 |
| Stable-Diffusion | 0.16 | 0.24 | 0.24 | 0.68 | 0.08 | 0.46 | 0.92 |
| Re-Imagen (K=2) | **0.80** | **0.80** | **0.82** | **0.92** | **0.54** | **0.80** | 0.98 |

Table 2: Human evaluation results for different models on different types of entities.

The inference speed is 30-40 secs for 4 images on 4 TPU-v4 chips. We demonstrate our human evaluation results for faithfulness and photorealism in Table 2.

We can observe that Re-Imagen can in general achieve much higher faithfulness than the existing models while maintaining similar photorealism scores. When comparing with our backbone Imagen, we see the faithfulness score improves by around 40%, which indicates that our model is paying attention to the retrieved knowledge and assimilating it into the generation process.

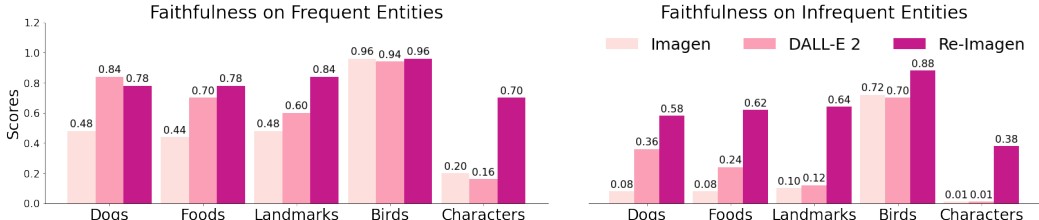

Figure 5: The human evaluation scores for both frequent and infrequent entities.

We further partition the entities into 'frequent' and 'infrequent' categories based on their frequency (top 50% as 'frequent'). We plot the faithfulness score for 'frequent' and 'infrequent' separately in Figure 5. We can see that Re-Imagen is less sensitive to the frequency of the input entities than the other models with only minor performance drops. This study reflects the effectiveness of text-to-image generation models on long-tail entities. More generation examples are shown in Appendix F.

## 4.3 ANALYSIS

**Comparison to Other Models** We demonstrate some examples from different models in Figure 6. As can be seen, the images generated from Re-Imagen strike a good balance between text alignment and entity fidelity. Unlike image editing to perform in-place modification, Re-Imagen can transform the neighbor entities both geometrically and semantically according to the text guidance. As a concrete example, Re-Imagen generates the *Braque Saint-Germain* (2nd row in Figure 6) on the grass, in a different viewpoint from to the reference image.

**Impact of Number of Retrievals** The number of retrievals $K$ is an important factor for Re-Imagen. We vary the number of $K$ for all three datasets to understand their impact on the model performance. From Figure 7, we found that on COCO and WikiImages, increasing K from 1 to 4 does not lead to many changes in the FID score. However, on EntityDrawBench, increasing K will dramatically improve the faithfulness of generated image. It indicates the importance of having multiple images to help Re-Imagen ground on the visual entity. We provide visual examples in Appendix D.

**Text and Entity Faithfulness Trade-offs** In our experiments, we found that there is a trade-off between faithefulness to the text prompt and faithfulness to the retrieved entity images. Based on Equation 6, by adjusting $\eta$, *i.e.* the proportion of $\hat{\epsilon}_p$ and $\hat{\epsilon}_n$ in the sampling schedule, we can control Re-Imagen so as to generate images that explore this tradeoff: decreasing $\eta$ will increase the entity's entity faithfulness but decrease the text alignment. We found that having $\eta$ around 0.5 is usually a 'sweet spot' that balances both conditions.

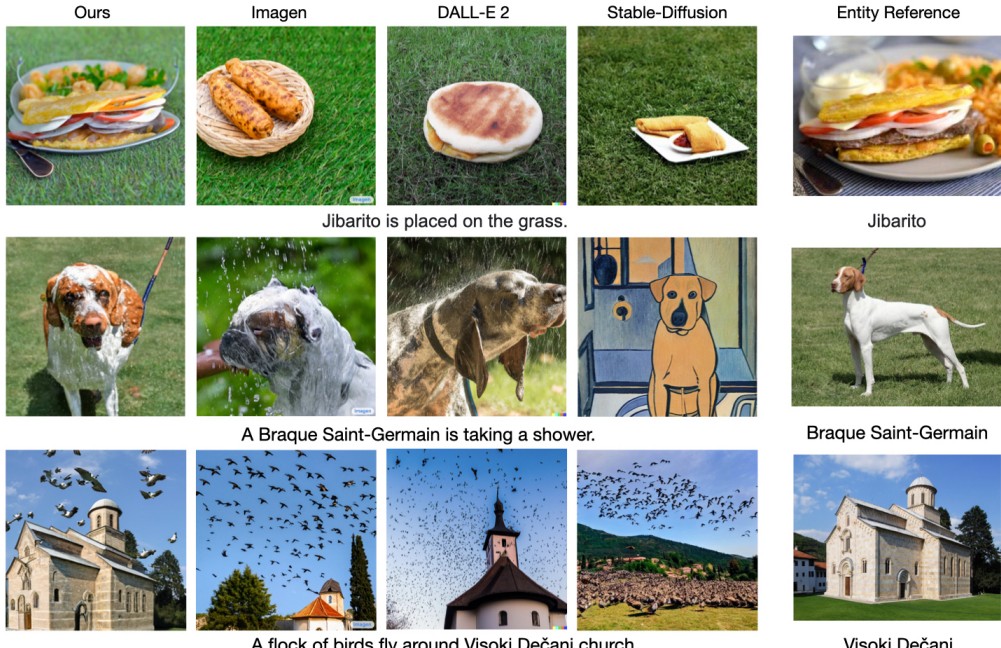

Figure 6: None-cherry picked examples from EntityDrawBench for different models.

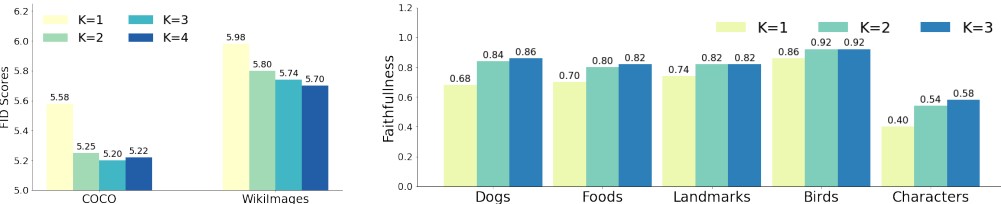

Figure 7: Ablation Study of retrieval number K on different datasets.

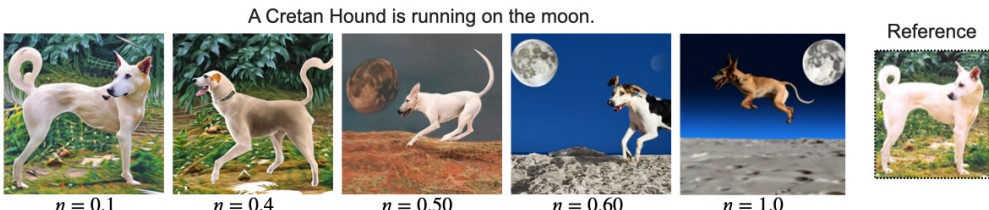

Figure 8: Ablation study of interleaved guidance ratio $\eta$ to show the trade-off.

## 5 CONCLUSIONS

We present Re-Imagen, a retrieval-augmented diffusion model, and demonstrate its effectiveness in generating realistic and faithful images. We exhibit such advantages not only through automatic FID measures on standard benchmarks (*i.e.*, COCO and WikiImage) but also through human evaluation of the newly introduced EntityDrawBench. We further demonstrate that our model is particularly effective in generating an image from text that mentions rare entities.

Re-Imagen still suffers from well-known issues in text-to-image generation, which we review below in section 5. In addition, Re-Imagen also has some unique limitations due to the retrieval-augmented modeling. First, because Re-Imagen is sensitive to retrieved image-text pairs it is conditioned on when the retrieved image is of low quality, there will be a negative influence on the generated image. Second, Re-Imagen sometimes still fails to generate high-quality images with highly compositional prompts, where multiple entities are involved. Thirdly, the super-resolution model is still not competent at capturing low-level details of retrieved entities leading to visual distortion. In future work, we plan to further investigate the above limitations and address them.

## ETHICS STATEMENT

Strong text-to-image generation models, *i.e.*, Imagen (Saharia et al., 2022) and Parti (Yu et al., 2022), raise ethical challenges along dimensions such as the *social bias*. Re-Imagen is exposed to the same challenges, as we employed Web-scale datasets that are similar to the prior models.

The retrieval-augmented modeling techniques of Re-Imagen have substantially improved the controllability and attribution of the generated image. Like many basic research topics, this additional control could be used for beneficial or harmful purposes. One obvious danger is that Re-Imagen (or similar models) could be used for malicious purposes like spreading misinformation, *e.g.,* by producing realistic images of specific people in misleading visual contexts. On the other side, additional control has many potential benefits. One general benefit is that Re-Imagen can reduce hallucination and increase the faithfulness of the generated image to the user's intent. Another benefit is that the ability to work with tail entities makes the model more useful for minorities and other users in smaller communities: for example, Re-Imagen is more effective at generating images of landmarks famous in smaller communities or cultures and generating images of indigenous foods and cultural artifacts. We argue that this model can help decrease the frequency-caused bias in current neural network-based AI systems.

Considering such potential threats to the public, we will be cautious about code and API release. In future work, we will explore a framework for responsible use that balances the value of external auditing of research with the risks of unrestricted open access, allowing this work to be used in a safe and beneficial way.

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

## A EXTENDED LITERATURE REVIEW

As aforementioned in the main text, in this section, we provide an additional review of related works, on (1) Retrieval-augmented Generative Models; and (2) Text-guided Image Editing.

**Retrieval-Augmented Generative Models**   Knowledge grounding has also drawn significant attention in the natural language processing (NLP) community. Different semi-parametric models like KNN-LM (Khandelwal et al., 2019), RAG (Lewis et al., 2020), REALM (Guu et al., 2020), RETRO (Borgeaud et al., 2021) have been proposed to leverage external textual knowledge into the transformer language models. These models have demonstrated great advantages in increasing the language model's faithfulness and reducing the computation/memory cost. Such attempts have also been made in visual tasks like image recognition (Long et al., 2022), 2-D scene reconstruction (Siddiqui et al., 2021), and image inpainting (Xu et al., 2021). Our proposed method follows the same theme to incorporate visual knowledge into a pre-trained text-to-image generation model to help the model generalize to long-tail entities or even unseen entities without scaling up the parameters.

**Text-Guided Image Editing**   The work of text-guided image editing aims at preserving the object's appearance while changing certain contexts in the image. Previously, GANs (Goodfellow et al., 2014) have been used to achieve significant performance on image editing (Zhu et al., 2016; Abdal et al., 2019; Zhu et al., 2020; Roich et al., 2021; Tov et al., 2021; Wang et al., 2022; Alaluf et al., 2022). The problem is also known as inversion as it normally requires finding the initial noise vector added in the generation process. More recently, Prompt-to-Prompt (Hertz et al., 2022) propose to use pre-trained text-image models for image editing. Image editing is focused on performing in-place modifications to the input image, either changing the global styles or editing a local region specifically without modifying the object's appearance. However, we treat the retrieved image as 'knowledge' and ground on it to synthesize new images. Thus, we are not restricted to in-place modifications and are able to perform more sophisticated transformations over the objects.

## B WIKIIMAGES DATASET

The WikiImages dataset is taken from WebQA (Chang et al., 2022). Images were crawled from Wikimedia Commons via the Bing Visual Search API. Since lots of Wikimedia's topics are not visually interesting, the authors seeded with natural scenes and gradually refine the search pool to obtain more interesting images. The images are mostly containing entities from Wikipedia or WikiData. However, the original dataset still contains heavy noises. Therefore, we apply further filtering to obtain the more plausible ones for image generation. Specifically, we remove all the image-text pairs with text lengths larger than 15 tokens and all the text with a date or wiki-id information.

## C  ENTITYDRAWBENCH

For dog breeds and birds, we sample 50 from Wikipedia Commons[3] as our candidates. For landmarks, we sample 50 from Google Landmarks (Weyand et al., 2020) as our candidate. For foods, we sample 50 from Wikipedia[4] as our candidates. For film characters, we collected 50 images from Starwars from Fandom[5]. We use appropriately paired source images as the retrieved 'knowledge'. For each entity category, we write 5 prompt templates with an entity name placeholder, which describes the entity in different scenes. Each entity will sample a template and replace the placeholder with the entity's name to generate a prompt, which is used as input to the text-to-image generation model.

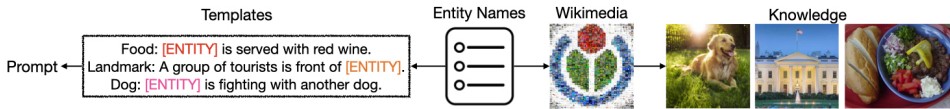

Figure 9: The construction process of EntityDrawBench. We first list entity names and then find their source images from Wikimedia, and finally generate prompts related to these entities.

We list all the prompt templates as Figure 10.

| Type | Template 1 | Template 2 | Template 3 | Template 4 | Template 5 |
|---|---|---|---|---|---|
| **Dog** | [DOG] is sleeping on the ground. | [DOG] is running by the river. | [DOG] is catching a frisbee. | [DOG] is taking a shower. | [DOG] is fighting with another dog. |
| **Food** | [FOOD] is placed on the grass. | [FOOD] is served with wine. | [FOOD] with popcorn on the side. | A dog is beside [FOOD] (food). | [FOOD] is decorated with flowers. |
| **Landmark** | A dog is sitting in front of [LANDMARK] | A big crowd of tourists in front of [LANDMARK]. | A rainy day in [LANDMARK]. | [LANDMARK] is lit up during the night. | cars parking in front of [LANDMARK]. |
| **Bird** | A [BIRD] is docking on a pier. | A [BIRD] is drinking water. | A [BIRD] is flapping its wings. | A [BIRD] is diving from the sky. | A [BIRD] is swimming in the river. |
| **Character** | The StarWars character [ENTITY] is flying in the sky. | The StarWars character [ENTITY] is standing in the water. | The StarWars character [ENTITY] is standing in the garden. | The character [ENTITY] is in a shopping mall. | The StarWars character [ENTITY] is in the kitchen. |

Figure 10: The EntityDrawBench prompt templates for all the different entity categories.

---

[3] https://commons.wikimedia.org/wiki/List_of_dog_breeds
[4] https://en.wikipedia.org/wiki/List_of_cuisines
[5] https://starwars.fandom.com/wiki/Main_Page

# D   IMPACT OF RETRIEVAL NUMBER K

We change the retrieval number K from 1 to 2 to see its impact on the model output. We show some examples in Figure 11 to demonstrate the advantage of having multiple retrievals to help the model better capture the visual appearance of the given entities.

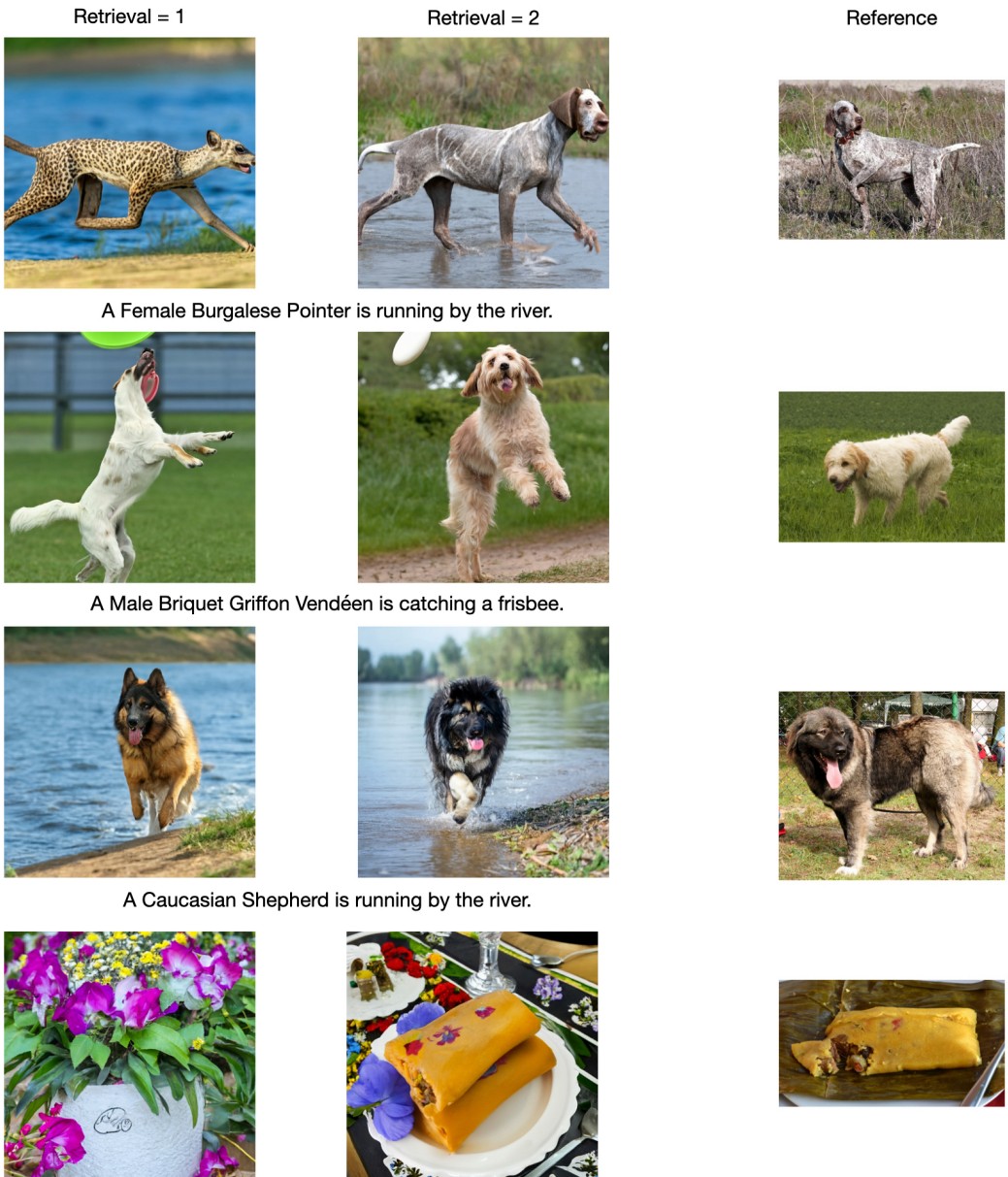

Figure 11: Generation Examples for setting K to 1 and 2.

# E CLASSIFIER GUIDANCE-FREE SAMPLING STRATEGY

We demonstrate different types of sampling strategy to leverage two conditions: standard joint condition guidance sampling, weighted guidance sampling, and our proposed interleaved guidance sampling.

The standard joint condition guidance only considers the joint diffusion score $\epsilon(x_t, c_n, c_p)$ to meet both conditions. In contrast, weighted guidance sampling uses the weighted sum of text-enhanced epsilon $\hat{\epsilon}_p$ and neighbor-enhanced epsilon $\hat{\epsilon}_n$. Our interleaved classifier guidance switches between $\hat{\epsilon}_p$ and $\hat{\epsilon}_n$, with a ratio of $\eta : 1 - \eta$. We plot their conceptual difference in Figure 12. Essentially, $\epsilon_n$ and $\epsilon_p$ do not have dependency in weighted sampling, however, they are dependent in interleaved sampling. In an extreme case where $\epsilon_n$ and $\epsilon_p$ are contradictory to each other, the model will get stuck in a local region. In contrast, Interleaved sampling can alleviate this issue.

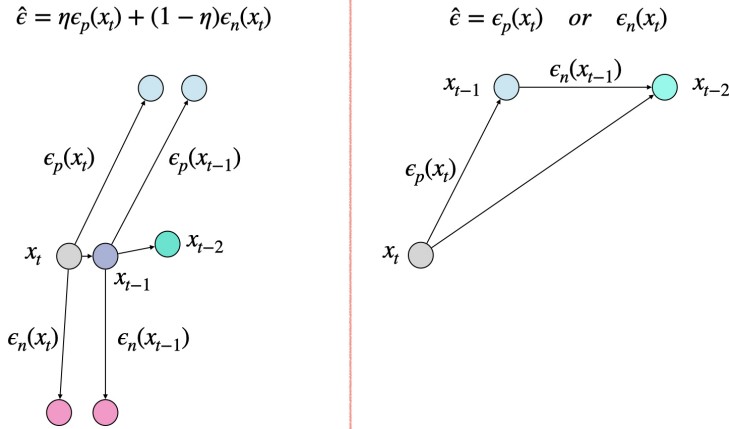

Figure 12: Weighted Guidance Sampling vs. Interleaved Guidance Sampling.

We compare 20 dog images generated from these three sampling strategies in EntityDrawBench. We vary the number of diffusion step to observe their human evaluation score curve and show case some generated outputs in Figure 13. As can be seen, the joint decoding is either dominated by the retrieval image or by the text prompt. Weighted and Interleave can help balance the two conditions to generate better images. We also found that with less sampling steps K=200, "weighted" sampling actually achieves better results than "interleaved" sampling. However, as the sampling steps increase, our proposed "interleaved" sampling achieves better human evaluation score.

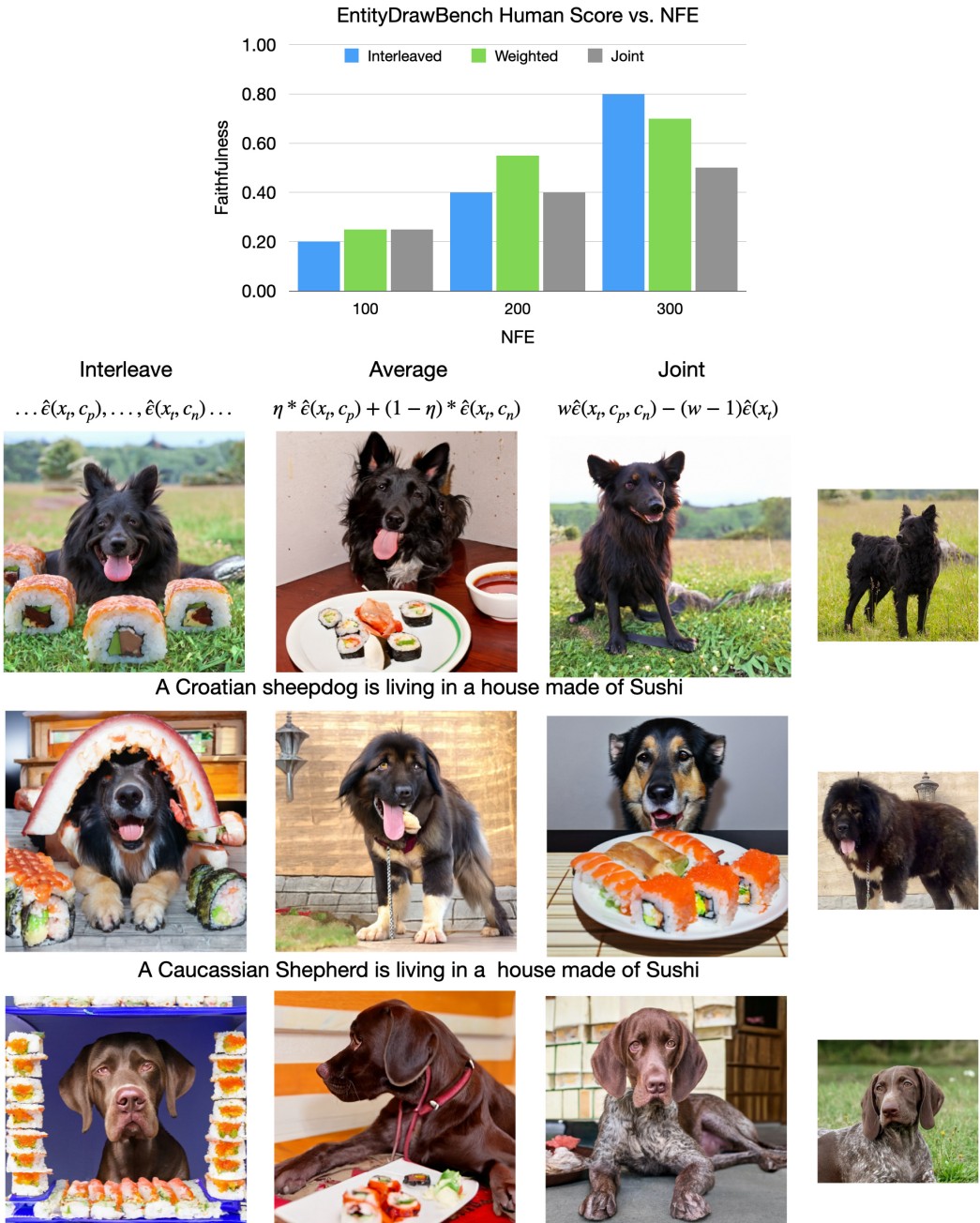

Figure 13: Different classifier-free guidance sampling strategy (Interleave is ours).

# F    GENERATION EXAMPLES

We provide more generation examples in Figure 14 and Figure 15.

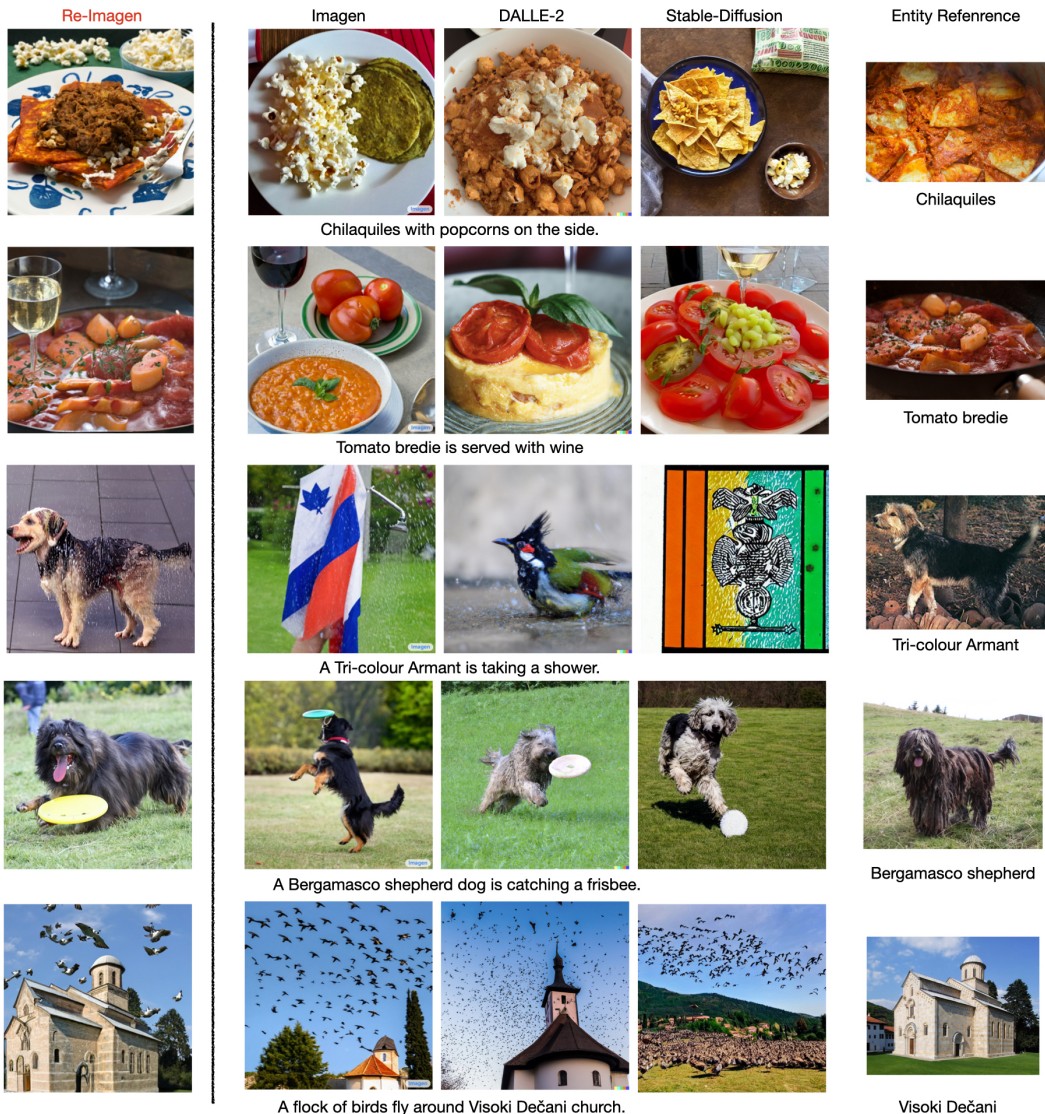

Figure 14: Extra None-cherry picked examples from EntityDrawBench for different models.

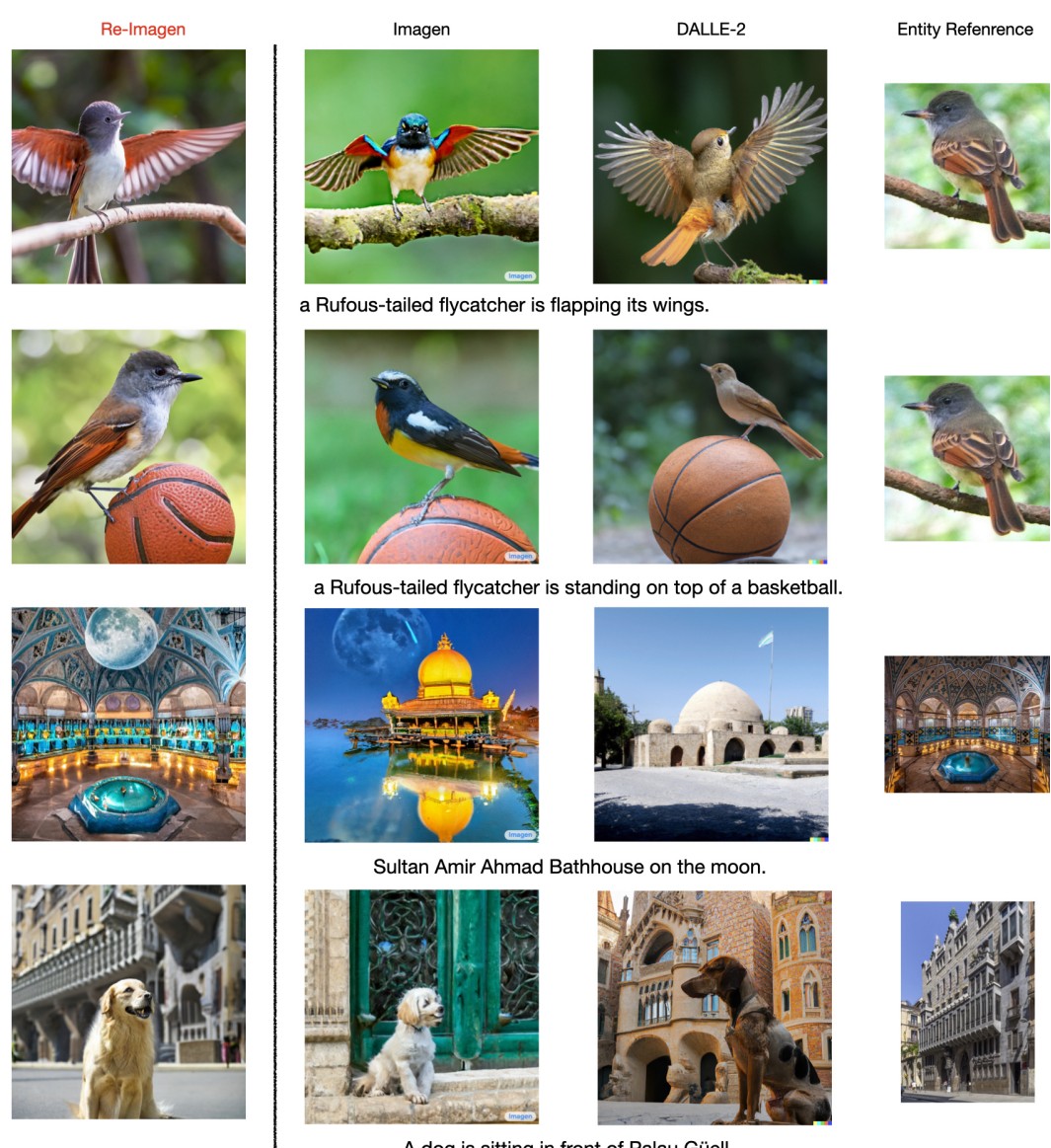

Figure 15: Extra None-cherry picked examples from EntityDrawBench for different models.

# G   IMAGINARY EXAMPLES

We provide generation results for imaginary scene in Figure 16.

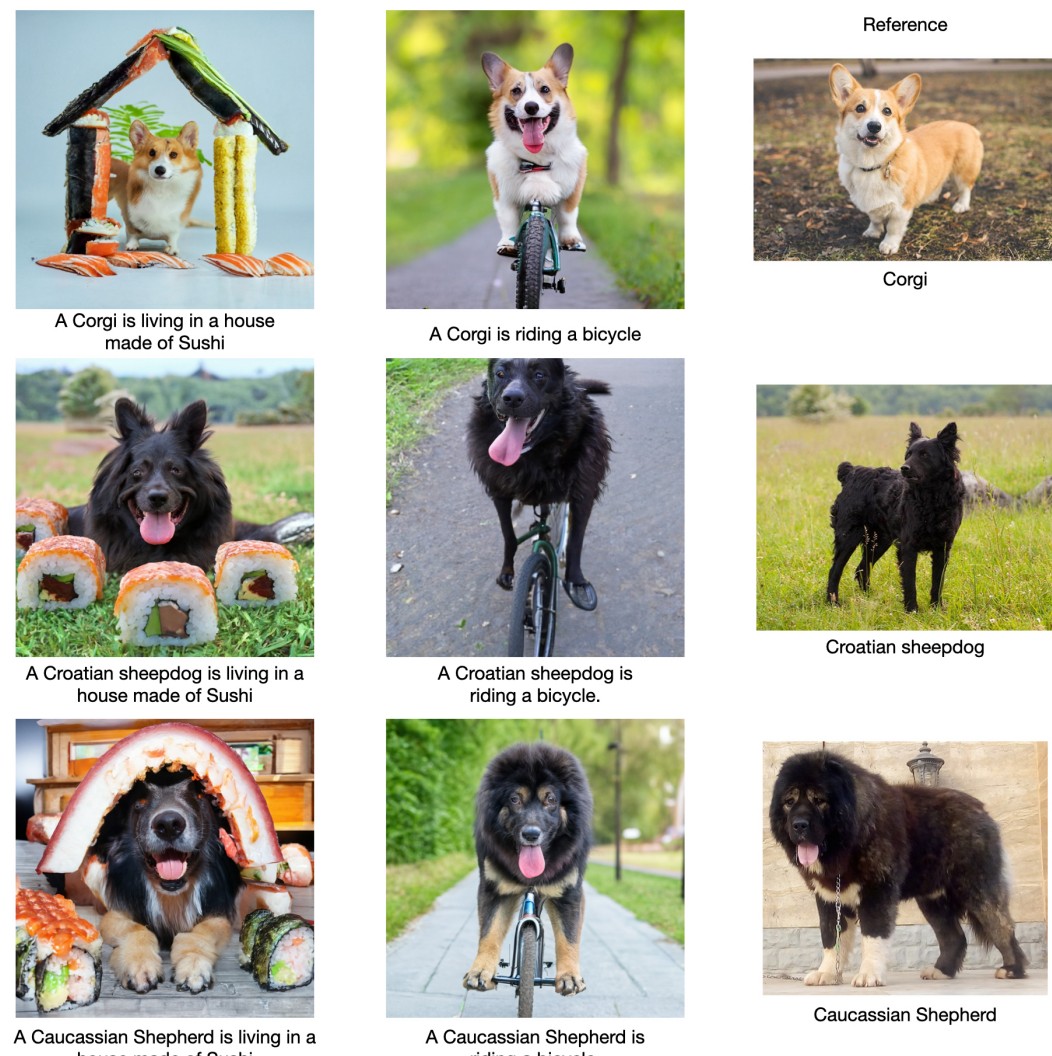

Figure 16: Imaginary Scenes generated by Re-Imagen.

## H    COMPARISON WITH DREAMBOOTH

We also add comparison to DreamBooth (Ruiz et al., 2022). We adopt almost the same input images from DreamBooth and display our generation results in Figure 17,  Figure 18 and  Figure 19.

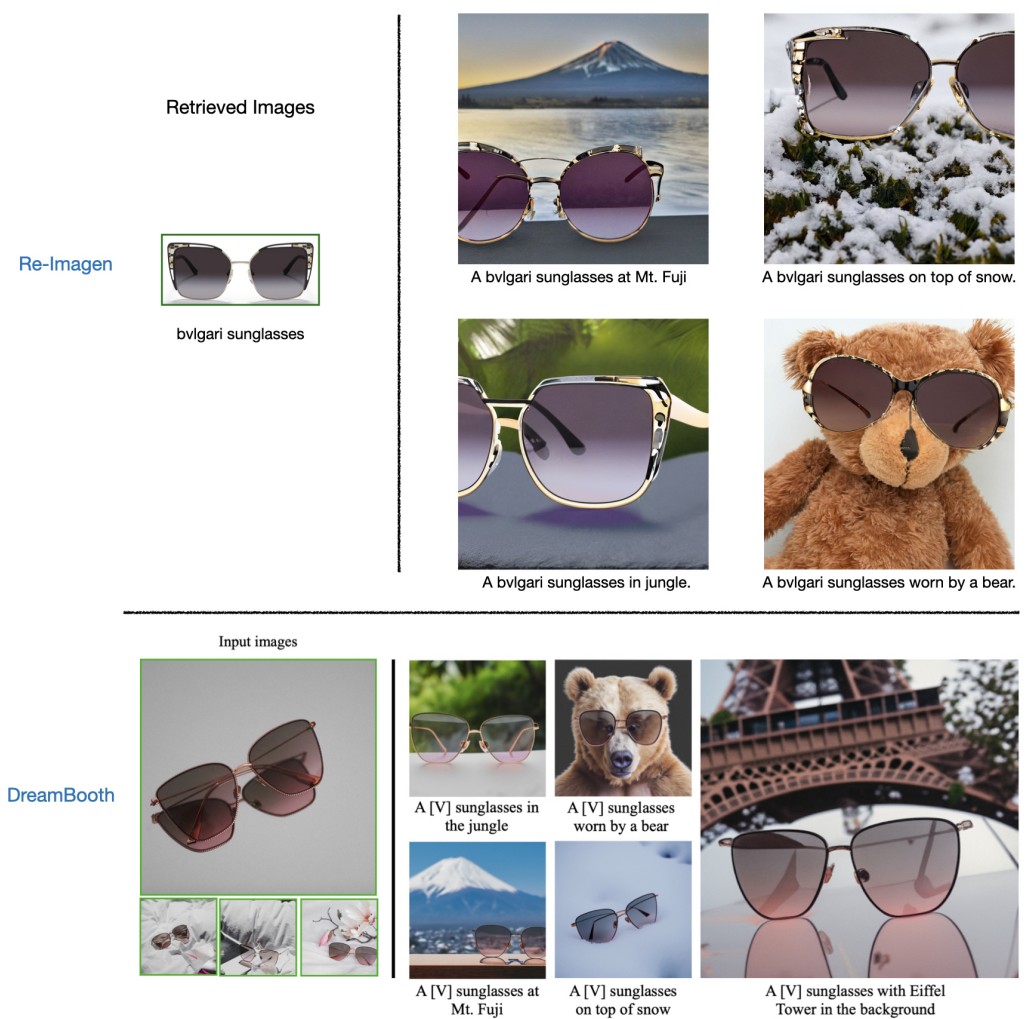

Figure 17: Imaginary Scenes generated by Re-Imagen.

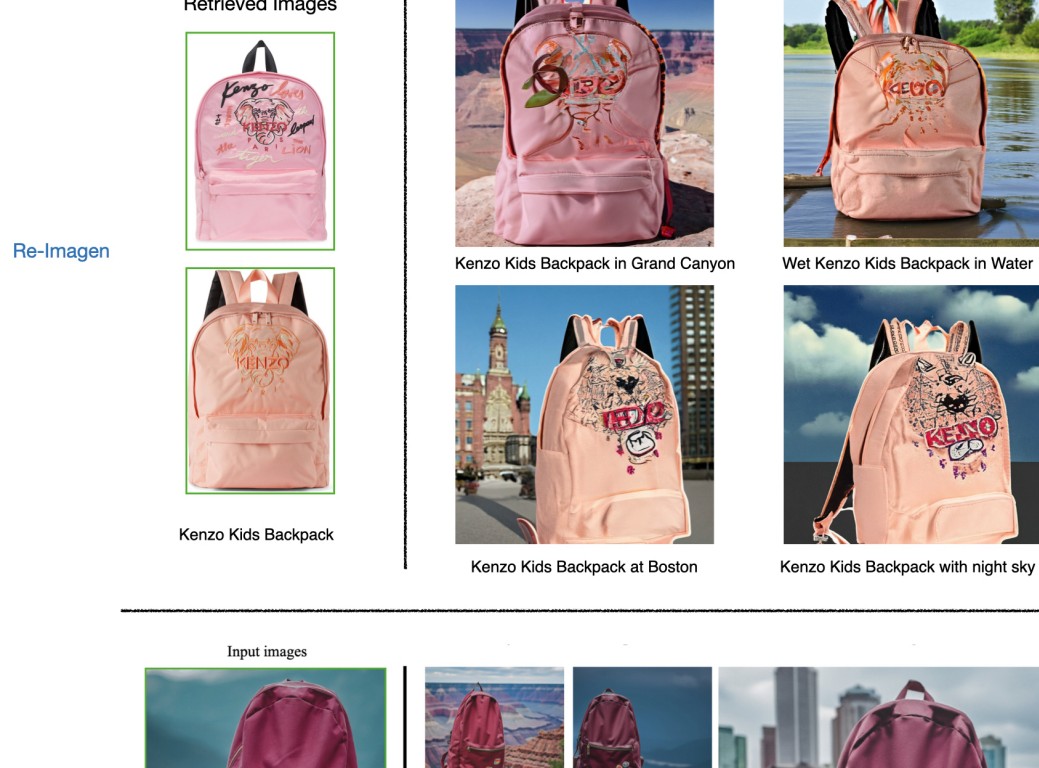

Figure 18: Imaginary Scenes generated by Re-Imagen.

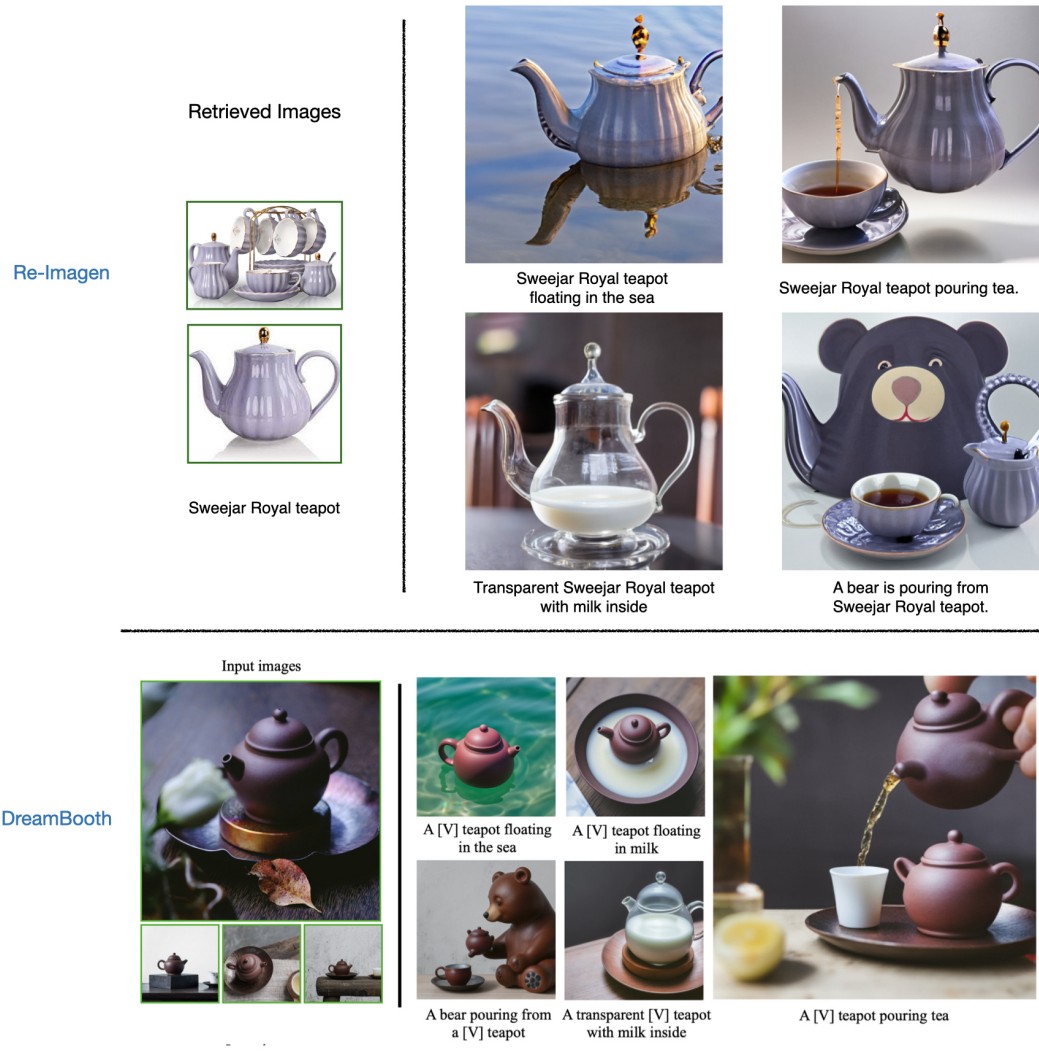

Figure 19: Imaginary Scenes generated by Re-Imagen.

## I  FAILURE EXAMPLES

We found that Re-Imagen can also fail in a lot of cases. We demonstrate a few examples in Figure 20. As can be seen, the model sometimes has a few failure modes: (1) the text input prior is too strong like 'Zoom' will be interpreted as a 'Zoom-in' picture by the model. (2) the model cannot ground the retrieval text on the retrieval image, for example, the model believes that only the 'beef tenderloin inside the bowl' is 'Escudella' rather than the whole stew, therefore generating 'beef tenderloin on the grass'. (3) the model can sometimes mess up two conditions, for example, the reference 'Australian Pinscher' and the 'rabbit' in the prompt gets mixed into a single object.

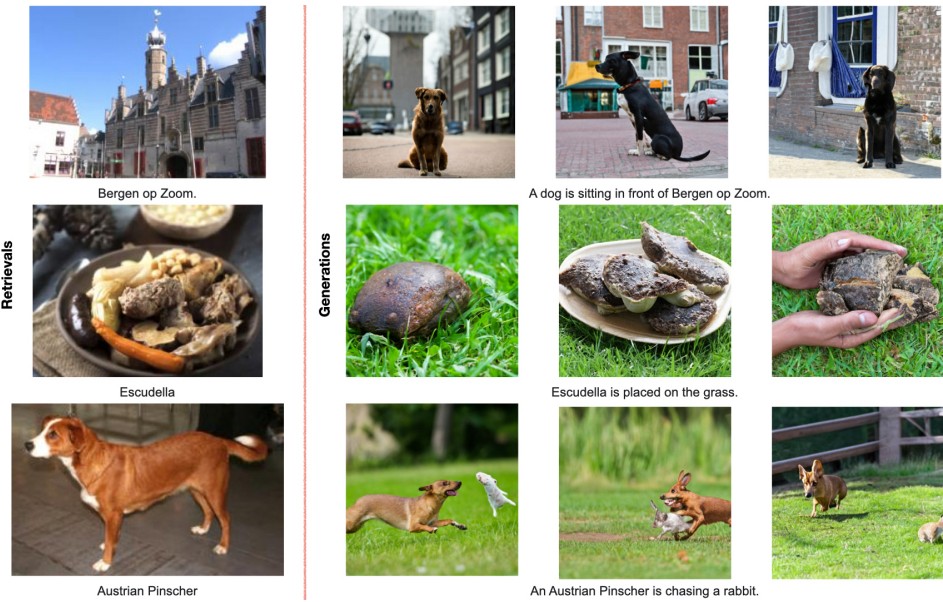

Figure 20: Failure examples from EntityDrawBench for dogs, landmarks, and foods.

