# OpenReview forum: "Re-Imagen: Retrieval-Augmented Text-to-Image Generator"
_ICLR.cc/2023/Conference — ICLR 2023 poster_

### Official Review · Reviewer_oSTW · 2022-10-23

**Confidence:** 5
**Correctness:** 3
**Technical Novelty And Significance:** 3
**Empirical Novelty And Significance:** 2
**Recommendation:** 5

**Clarity, Quality, Novelty And Reproducibility:**

Clarity: The clarity of this paper is good. The pipeline of the proposed methods is clear and the illustrations are pretty helpful for the readers although there are still lots of typos in the paper.

Quality:The completion of this paper is still good. However the comparisons in the experiments may be not fair. The experiment results especially the FID score cannot support the effectiveness  of the proposed methods.

Novelty: The novelty of this paper is limited. Although  this paper introduces a new challenging benchmark for text-to-image generaiton, the methods of  this paper are not very novel. The Interleaved Classifier-free Guidance seems an implementation trick for the multiple conditions for diffusion models.

Reproducibility: The clarify of the proposed methods are clear. It is possible to reproduce the performances of the proposed methods but it may be very challenging due to the lackness of the key codes and details.

**Strength And Weaknesses:**

Strength:
1) This paper introduces an interesting viewpoint for text-to-image generation by involving the retrieval-based images during the generation procedure.
2) The proposed methods achive good performances on the datasets and the newly-introduced benchmark may be helpful for the text-to-image task.

Weakness:
1) The comparison is not fair in the experiments. As for the most of the previous methods, the images are generated from the only text inputs. While, the proposed methods involve the images retrieved from texts by the datasets, which will make a big difference on the evaluation of the generated images, especially the FID scores.
2) The novelty is limited. Although the retrieval-augmented methods achieve good performances and provide another viewpoint for image-guided text-to-image generation, the multiple conditions for diffusion models are not novel. The UNet architecture is also widely used in the diffusion models.

**Summary Of The Paper:**

This proposes a retrieval-augmented text-to-image model Re-Imagen for text and image guided image generation. The proposed method Re-Imagen interleaves classifier-free guidance during sampling to ensure both text alignment and entity fidelity and achieve good performances on the two datasets. This paper also introduces a more challenging benchmark EntityDrawBench for the text-to-image generation.

**Summary Of The Review:**

This paper introduces a novel retrieval-augmented text-to-image model and a more challenging benchmark for text-to-image generation. The retrieval-augmented methods may be helpful for the application for the text-to-image generation. However, the retrieved image involving in the image generation procedure makes it not fair to be compared with the previous methods which generates the images only from the texts. The experiements cannot support to prove the effectiveness of the proposed methods. Futhermore, the novelty of the proposed methods is limited. This paper is not good enough for publication.

---

> ### Author Response · Authors · 2022-11-12
> **Response**
>
> Thank your so much for your constructive feedback.
>
> * Fair Comparison
>
> We not only compare with text-to-image models like DALLE-2, Imagen, etc but also compare with other retrieval-augmented models like KNN-Diffusion (Ashual et al.) and Memory-Driven Text-to-Image (Li et al.). We restructure Table 1 to make this more clear. These two retrieval-augmented models use the same setting as ours where the model accesses an additional database to help generation. So it's a completely fair comparison. According to Table 1 in the revision version, Re-Imagen can beat them by 11 FID on COCO. We think these strong results reflect the effectiveness of our model in leveraging additional retrievals.
>
> * Regarding FID score
>
> Optimizing FID score is not the only purpose of this paper, nor is it a sufficient metric to reflect the "faithfulness/fidelity" of generated images. That’s why we did more experiments on EntityDrawBench (now expanded to six categories) and uses human evaluation to show the faithfulness of images generated by Re-Imagen.
>
> * Novelty of Interleaved Classifier-free Guidance
>
> Interleaved guidance is not heuristic. Let $z$ be data $c_p$ be the text condition and $c_n$ be the neighbor condition. The interleaved guidance sampling approximately aims to maximize the objective of $\log( p(z|c_p,c_n)p^w(c_p|z, c_n)p^w(c_n|z, c_p))$. During training, we randomly mask out conditions $c_p$ or $c_n$ to help the model learn marginalized data distribution $p(z|c_p) = \int_{c_n} p(z, c_n|c_p)$ and $p(z|c_n) = \int_{c_p} p(z, c_p|c_n)$  through diffusion model. Then we adopt the Bayesian rule to compute $p(c_p|z, c_n) \propto \frac{p(z|c_p, c_n)}{p(z|c_p)}$ and $p(c_n|z, c_p) \propto \frac{p(z|c_n, c_p)}{p(z|c_n)}$ as implicit classifier to provide disentangled guidance.
>
> **We also conduct in-depth comparison between different types of sampling strategies in Appendix E.**
>
> * Novelty Limitation
>
> We believe our paper has novelty in the following aspects:
>
> (1) Our model is the first (or among the first few) to leverage retrievals into large-scale diffusion model training to enhance the faithfulness of image generation.
>
> (2) We design the new interleaved sampling strategy to ensure model generation conforms with both conditions.
>
> (3) Our model obtains very strong performance and beats the other retrieval-augmented image generation models like KNN-Diffusion by at least 11 FID on COCO.
>
> (4) We create the new EntityDrawnBench dataset and show the effectiveness of our model on this dataset, especially in generating long-tail entities.
>
> (5) Our model has a far-reaching impact: in general, the model can help improve the trustworthiness of the current model by providing more attribution. On the other hand, the model is a big step to reduce the bias presented in image-generation models. For example, Re-Imagen can generate visual objects associated with underrepresented cultures or areas. (We showed plenty of examples in the Appendix.)

---

### Official Review · Reviewer_UC3H · 2022-10-24

**Confidence:** 4
**Correctness:** 3
**Technical Novelty And Significance:** 3
**Empirical Novelty And Significance:** 3
**Recommendation:** 6

**Clarity, Quality, Novelty And Reproducibility:**

Clarity: The ideas in the paper are reasonably well conveyed. However, as I pointed above, more discussions on design choices might have been nice.

Quality: This is a good quality work. Results are quite strong.

Novelty: The idea of using retrieval in diffusion models is not something new. There are many concurrent papers as authors point out. However, showing these impressive results at this scale is indeed a contribution, one that is valuable to the community.

Reproducibility: This is something I am a bit concerned about. Many of the ablations and design choice discussions are missing. The authors did provide other details, but adding some more detailed discussions would help with reproducibility.


**Strength And Weaknesses:**

Strengths:

The use of external knowledge base to augment the diffusion models is intuitive and a powerful idea. It is very hard to encode the entire world knowledge in a 2.5B parameter model, hence, these models are bound to fail in rare modalities. External knowledge base comes to rescue in this case.

The idea proposed by authors is simple, yet quite effective (as shown in  experimental studies). This model can also be finetuned from the original text-to-image model (Imagen in this case), which is something really nice. This way, a lot of compute can be saved.

The modification to the existing classifier free guidance to have optional conditioning on both nearest neighbor image features and text is also interesting.

The experimental results are strong. Both on zero-shot FID and human evaluation, the authors achieve SOTA.

Weakness:

I have a few comments, mainly with respect to some of the design choices made. I feel like the authors mainly mentioned these choices, and did not have a good discussion / justification.

On the choice of number of nearest neighbors during training: The authors just mention that they use k=2. But why just k=2? I can imagine that as k increases, the number of queries and keys in the cross attention would increase, hence the computational burden would go high. Is that the reason why k=2 was chosen? In a ideal case, performing an ablation on k would be good. But I can understand that ablations might be too expensive in these models. Hence, having a discussion on why this decision was made and what considerations were taking into account to end up in this decision would help the research community.
On the choice of number of nearest neighbors during inference: Should the same k be used during training and inference? While it might be true that performing an ablation on k during training might be infeasible, performing an ablation on k during inference might not be that hard. It would be great if authors had done that. K is an important parameter in retrieval, hence understanding the sensitivity of k might be beneficial.
Modified classifier-free guidance: The modification to the classifier free guidance makes sense to me. In the implementation, the authors randomly flip a coin and choose one of the two scores - text score and neighbor score in reverse diffusion. Why should this be done? Why not use a weighted combination of both scores. That is, why not use $\eta \hat{\eps}_{p} + (1-\eta)\hat{\eps}_{n}$ as the score? This might increase the NFE’s, but it might be worthwhile to study if this gives any gains. An ideal experiment would be to plot NFE-FID tradeoff for using this combined score vs using the proposed algorithm.
I would have liked it if authors provided more visualizations. One of the power of techniques like Imagen is the ability to image zero-shot content. For instance, “Corgi lives in a house made out of sushi”. But none of such visualizations is provided in this paper. I am wondering if this ability got worse due to the nearest neighbor finetuning because such samples are not present in retrieval dataset. COCO FID might not measure this as captions in COCO are not containing such novel texts. A discussion on this might be beneficial.
A discussion on computational overhead: This is related to 1, but I am curious about what the additional computational overhead is because of the added attention tokens?
BM-25 seems to be a very simple retrieval algorithm. Will you get more gains by using CLIP text?


**Summary Of The Paper:**

This paper proposes a retrieval augmented diffusion model for the task of text to image synthesis. The idea is to condition the diffusion model on the external knowledge base in addition to the text embeddings that are used in text-to-image diffusion models. Conditioning on the retrieved nearest neighbor embeddings happens using the cross-attention layer. During inference, the model retrieves the image-text pairs corresponding to the input text prompt and then samples the result. Experiments show SOTA results on zero-shot FID on COCO, other benchmark datasets and human evaluation.

**Summary Of The Review:**

I think the paper is a good contribution to the community since it presents good results. However, as I pointed above, some of the ablations, discussions and visualizations are missing. Adding these would have made it a very strong submission.

---

> ### Author Response · Authors · 2022-11-12
> **Response**
>
> Thank your so much for your constructive feedback.
>
> * why just k=2 during training
>
> Your thoughts are correct. The bottleneck here is the memory cost. Since the retrieved neighbors are reusing the downstack encoder, increasing K would lead to higher memory consumption. We have tried increasing K=3, but the batch size needs to be halved to accommodate the model. In the end, K=3 does not provide much gain over K=2. However, having K=2 is better than K=1 because the model learns to deal with multiple retrieval inputs.
>
> * Impact of K during inference
>
> The attention mechanism is flexible, which allows as many retrieved image-text pairs as the memory would accommodate. **We provided some ablation studies  regarding K in Figure 7 to see its impact on the FID score and human evaluation score.** We found that increasing K during inference does not give much gain on FID metrics, but it gives significant boost on image faithfulness with human evaluation. The model tends to capture the entity’s visual details better by referring to multiple images describing the same entity.
>
> * Modified classifier-free guidance
>
> **We implemented your suggested “weighted combination” version for epsilon prediction and made some conceptual and empirical comparisons in Appendix E**. In essence, weighted combination assumes weaker dependency between $\epsilon_p$ and  $\epsilon_n$, while interleaved sampling enforces a stronger dependency between  $\epsilon_p$ and  $\epsilon_n$. We plot trade-off curves of NFE-Human_Score. We found that weighted combination achieves better score with fewer diffusion steps, but falls behind interleaved sampling when diffusion steps increase. It roughly conforms with our expectations.
>
> **We also plotted the outputs from different sampling strategies in Appendix E to provide better visualization.**
>
> * Imaginary Scenes
>
> We show some examples like these in the Appendix G and H to confirm that Re-Imagine can handle these prompts relatively well.
>
> * Computation Overhead
>
> We have an earlier version implementing CLIP retriever. Our preliminary result was that using CLIP as the retriever does not help much in terms of FID score. But we did not fully test on human evaluation. Due to the time constraint, we are not able to finish that during the rebuttal, but we are happy to conduct more experiments later.
>
> * Reproducibility and API release
>
> This is one of our high priority tasks. We are trying to find the safest way to release the model to prevent any potential ethical and copyright issues. There has been progress regarding the public API release of our backbone Imagen model. Please refer to:
> https://www.dpreview.com/news/2363850384/google-releases-text-to-image-ai-model-imagen-for-public-use-for-the-first-time. We plan to follow a similar path to release APIs to verified users.
>
> On the other hand, we are already collaborating with external teams to implant this algorithm to stable diffusion. We are using LAION-aesthetics as the training dataset to tune stable diffusion models to ground on retrievals. We already have good progress now and expect to release it once it’s reaching expected performance.

---

### Official Review · Reviewer_xjWB · 2022-10-24

**Confidence:** 4
**Clarity, Quality, Novelty And Reproducibility:** 1. Will the code and dataset be publi…
**Correctness:** 3
**Technical Novelty And Significance:** 3
**Empirical Novelty And Significance:** 3
**Recommendation:** 6

**Strength And Weaknesses:**

**Strength**

1) The issue that this paper tried to solve is important and critical in nowadays text2image generation. The topic is meaningful.
2) This paper is well-written and easy to follow.
3) The qualitative results is compelling.

**Weaknesses**

1) Research topic definition changing.

The research topic is text-to-image generation, which is presumed to take text as input and image as output. However, though this work means to solve the problem of uncommon entities generation in text-to-image generation, it brings subtle but certainly important changing of the "research topic definition". An extra input of a set of knowledge base is needed for the inference of this model, which makes the problem to be like text (with knowledge base)-to-image generation, with more strong conditions. The changing of the definition mainly brings a problem that the usage bar for normal users is actually pulled up.

2) The comparisons on Table 1, Table 2 and Table 3 have some concern in fairness.

(a) For the comparion between Re-Imagen ( =BM25; B=COCO; k=2) to Imagen on Table 1, (i) the # of Params are different, (ii) the training set is different (iii) the metric FID-30K and Zero-shot FID-30K is different, it seems to be arbitrary and misleading to say: “Re-Imagen (with the COCO database) can achieve a significant gain on FID-30K without fine- tuning: roughly a 2.0 absolute FID improvement over Imagen.”.

(b) For the comparison on Table 2, the # of Params are different, it hard to deduce meaningful and solid conclusions. The factors should be disentangled to disclose the key insights and observations in the experiments.

(c) For the comparison on Table 3, the motivation why these three types of visual entities (dog breeds, landmarks, and foods) are picked may be clarified. A column of results of broader entities can be more convincing.




**Summary Of The Paper:**

This paper proposed an approach to deal with the problem, that state-of-the-art models often have difficulty generating images of uncommon entities. The main contributions are 1) a novel retrieval-augmented text-to-image model Re-Imagen. 2)  interleaved classifier-free guidance. 3) a benchmark.

**Summary Of The Review:**

This paper tried to solve an important problem, and the method is reasonable and works. The main concerns are in the fairness in the comparisons in experiments and the reimplementation of this method for both data and model.

---

> ### Author Response · Authors · 2022-11-12
> **Response**
>
> Thank your so much for your constructive feedback.
>
> * Research topic definition changing
>
> Our model only takes a user prompt as the input, **and uses a retriever to automatically search for relevant image-text pairs from a database.** Our experiments use LAION or WikiMedia or Google Internet Search to automatically find the most relevant “entity visual knowledge”. **There is no extra requirement or burden for the user at all. This retrieval step is hidden by the underlying system so the end user won’t notice it.** The inference speed is almost the same as Imagen since the retrieval can be done in milliseconds.
>
> From the model provider perspective, we do need extra cost to serve a database. But due to the maturity of current IR techniques, this is done very easily with existing libraries like FAISS, Pyserini, etc. If the provider does not have the memory space to serve it, they can relegate this to Google Search or Bing Search.
>
> With such a low-cost retrieval augmentation, the benefits are tremendous. It not only improves the attribution of generation process, but also significantly improves the faithfulness of generated images.
>
> * Fairness in Comparison
>
> (a.1) To address the concern w.r.t our additional number of parameters, we also included results for Re-Imagen-small with 2.4B, which continue to outperform Imagen-3B significantly on COCO (5.7 vs 7.2 FID) and WikiImages (6.04 vs 6.44).
>
> (a.2) **Our training set (Internal Image-text dataset) is a subset of Imagen’s training set**. Our retrieval database LAION is also a subset of Imagen's training data. No additional data is used for Re-Imagen at all.
>
> (b) **We added Re-Imagen-small’s results in Table 1**, a smaller variant that continues to outperform Imagen 3B on this dataset.
>
> (c) We chose these three categories to include the most common visual entities. We also avoid “human” categories  to avoid potential copyright issues or ethical concerns. In our revision, **we added “birds” and “characters” to enrich the EntityDrawBench** and display more results and find consistent performance gain.
>
> (d) **We added a “Broader” category to include random real-world objects like “teapot, backpacks, etc” and write creative prompts for these broader objects**. We report our results in Table 2 and show examples in Appendix H.
>
> * Will the code and dataset be public available? If yes, which parts of the code/data used will be released?
>
> This is one of our high priority tasks. We are trying to find the safest way to release the model to prevent any potential ethical and copyright issues. There has been progress regarding the public API release of our backbone Imagen model. Please refer to:
> https://www.dpreview.com/news/2363850384/google-releases-text-to-image-ai-model-imagen-for-public-use-for-the-first-time. We plan to follow a similar path to release APIs to verified users.
>
> On the other hand, we are already collaborating with external teams to implant this algorithm to stable diffusion. We are using LAION-aesthetics as the training dataset to tune stable diffusion models to ground on retrievals. We already have good progress now and expect to release it once it’s reaching expected performance.
>
> * Updates
>
> Besides, we would encourage you to check out the Change Log of the paper. We have added many experiments and visualization to help readers better understand Re-Imagen.

---

### Official Review · Reviewer_T7pn · 2022-10-24

**Confidence:** 4
**Correctness:** 3
**Technical Novelty And Significance:** 2
**Empirical Novelty And Significance:** Not applicable
**Recommendation:** 6

**Clarity, Quality, Novelty And Reproducibility:**

The paper writes clearly. Because of the simple idea, I believe it can be easily reimplemented. However, as far as I know, the behind diffusion model has not been released, which makes it hard for reproducibility and benefits the general research. The authors should justify how their works can contribute to general research without many computational resources properly.

**Strength And Weaknesses:**

Strengths:

+ The proposed idea of taking the power of retrieved images for generating better results is simple but effective. The effectiveness has also been demonstrated through the experiments.

+ I enjoy reading this paper, and it writes generally well, though with some typos (e.g., "on two dasets:").

+ This paper works on a recent hot topic, which is text-based image generation. The idea of customizing with more conditional images can push forward the development of this area.

+ I appreciate the limitations

Weaknesses:

+ Some parts of the model descriptions may not be clear. To my understanding, the Re-Imagen allows multiple retrieved images as inputs, but how exactly the model balances them or takes them as inputs? How is the model architecture designed? Does this mean the number of retrieved images should be the same as the training? What if the retrieved images share much diverse appearance, how the model handles such a case?

+ I feel the way of improving the classifier-free guidance is rather heuristic. Are there any insights that why the undesired imbalance happens?

+ Though the idea is effective, its underlying technical contribution is rather limited, as adding the retrieval images as another condition is straightforward for implementing the such retrieval-image-augment idea.

+ As an image generation work, I would like to see much more visual results than the paper and the supplementary document provided. It is encouraged to release a project webpage containing more results for demonstrating the generality of the proposed method.

+ Will the model be released publicly? If not, how this work can contribute to the research community is encourag

**Summary Of The Paper:**

This paper aims to utilize the retrieval image(s) for extending the model's ability for generating images. In this way, the model takes the text prompt as the input, by fusing with the noisy predicted by the retrieval images, it can generate the images that the dataset seldom contains. To balance the generation ability among text prompts and retrieved images, the paper also introduces a hand-crafted weighted scheme. Several experiments have been conducted to generate the efficiency of the method.

**Summary Of The Review:**

Generally speaking, the effective idea and good results make me lean toward acceptance. However, the rather hand-crafted designed combination way makes the method rather heuristic. The technical contribution is still limited. Thus, I can only put it a little bit over the boundary.

---

> ### Author Response · Authors · 2022-11-11
> **Response**
>
> Thank your so much for your constructive feedback.
>
> * Some parts of the model descriptions may not ... same as the training?
>
> We use an attention mechanism to integrate retrieval features into Re-Imagen. The attention mechanism is flexible, which allows as many retrieved image-text pairs as the TPU/GPU memory would accommodate. If multiple images are given, the attention module will decide which ones to focus on based on the attention score.
> During training, we use K=2 for the 2B model because it provides best trade-offs between computation cost and performance. With the computation budget fixed, increasing K further would require a smaller batch size. **During inference, the model does not need to strictly follow K=2, we did experiments with K=1,2,3,4 and reported our experimental results in Figure 7.** We found that increasing K during inference does not give much gain on FID metrics, but it gives significant boost on image faithfulness with human evaluation. The model tends to capture the entity’s visual details better by referring to multiple images describing the same entity.
>
> * What if the retrieved images share a diverse appearance, how does the model handle such a case?
>
> If multiple retrieved images have a shared object, like “Corgi in Kitchen”, “Corgi in Garden”, and “Corgi in river”. These retrievals can help the model better associate what Corgi’s visual appearance and thus improve model generalization.
> However, if the retrieved images are totally distinct from each other without any shared common theme. The model would just pick the most similar one (textual similarity or visual similarity) and ground on that for generation while ignoring the others.
>
> * I feel the way of improving the classifier-free guidance is rather heuristic. Are there any insights that why the undesired imbalance happens?
>
> Reason for imbalance: the two conditions “prompt” and “neighbors” are both provided as features to the diffusion model through the attention mechanism. It means that one of them could have significantly higher similarity to $x_t$, the diffusion model would pay most of its attention on this condition along the way while ignoring the other condition. Such behavior can harm the final performance significantly.
>
> Interleaved guidance is not heuristic. Let $z$ be data $c_p$ be the text condition and $c_n$ be the neighbor condition. The interleaved guidance sampling approximately aims to maximize the objective of $\log( p(z|c_p,c_n)p^w(c_p|z, c_n)p^w(c_n|z, c_p))$. During training, we randomly mask out conditions $c_p$ or $c_n$ to help the model learn marginalized data distribution $p(z|c_p) = \int_{c_n} p(z, c_n|c_p)$ and $p(z|c_n) = \int_{c_p} p(z, c_p|c_n)$  through diffusion model. Then we adopt the Bayesian rule to compute $p(c_p|z, c_n) \propto \frac{p(z|c_p, c_n)}{p(z|c_p)}$ and $p(c_n|z, c_p) \propto \frac{p(z|c_n, c_p)}{p(z|c_n)}$ as implicit classifier to provide disentangled guidance.
>
> **We compare different types of sampling strategies in Appendix E.**
>
> * Though the idea is effective, its underlying technical contribution is rather limited, as adding the retrieval images as another condition is straightforward for implementing the such retrieval-image-augment idea.
>
> The main idea is straightforward and intuitive. However, making this framework work requires non-trivial technical and engineering efforts. There are a series of design choices we need to make correctly  in order to achieve good performance like the choice of training corpus, retrieval function, model architecture, model sampling, and evaluation. Our final results are strong and compelling, which proves the effectiveness of this straightforward idea. On the other hand, we think being simple yet effective is indeed a strength rather than weakness.
>
> * As an image generation work, I would like to see much more visual results than the paper and the supplementary document provided. It is encouraged to release a project webpage containing more results for demonstrating the generality of the proposed method.
>
> We added a lot of image generation results in the Appendix. We are already working on releasing the project webpage. We will put more interactive examples on the webpage.

---

> > ### Author Response · Authors · 2022-11-11
> > **Respnose resumed.**
> >
> > * Will the model be released publicly? If not, how this work can contribute to the research community is encouraged.
> >
> > This is one of our high priority tasks. We are trying to find the safest way to release the model to prevent any potential ethical and copyright issues. There has been progress regarding the public API release of our backbone Imagen model. Please refer to
> > https://www.dpreview.com/news/2363850384/google-releases-text-to-image-ai-model-imagen-for-public-use-for-the-first-time for details. We plan to follow a similar path to release APIs to verified users.
> >
> > On the other hand, we are already collaborating with external teams to implant this algorithm to stable diffusion. We are using LAION-aesthetics as the training dataset to tune stable diffusion models to ground on retrievals. We already have good progress now and expect to release it once it’s reaching expected performance.

---

### Author Response · Authors · 2022-11-11
**Updated Version: Change Log**

We thank all the reviewers for their thoughtful comments. We have added many experiments and generation results to the revision based on these feedbacks. Here is the change log:
1. As suggested by Reviewer xjWB, we expand EntityDrawBench to include three more categories, i.e. birds, film characters. We found that our model is consistent on these new categories, the improvement on film character is even larger than the old categories.
2. As suggested by Reviewer xjWB, we also added broader to EntityDrawBench to include broader sets of objects like "vase, teapot, backpack, ...". We add comparisons to DreamBooth with similar inputs. We found that our generation is mostly on par with DreamBooth even without any object-specific fine-tuning. The comparisons are in Appendix H.
3. As suggested by Reviewer T7pn and UC3H, we add ablation study to study how K affects the model performance. It turns out that as K increases, though FID does not show significant boost, the human evaluation over the image faithfulness improves. With K=2, we increase the faithfulness score by almost 10%. We also compare the image outputs of K=1 vs K=2 in Appendix D.
4. As pointed by Reviewer xjWB and oSTW to make more fair comparisons. In Table 1, we include Re-Imagen-small, which has less parameters than Imagen for comparison. We found Re-Imagen-small can still beat Imagen significantly on COCO dataset.
5. To respond to Reviewer xjWB and oSTW on "research topic change",In Table 1, we also include other retrieval-augmented text-to-image generation models, which use the same setup as ours.  We found that Re-Imagen can beat KNN-diffusion and Memory-Drive T2I by 11 FID score. It suggests the effectiveness of our retrieval-augmented architecture.
5. We discuss other different types of classifier-free guidance as pointed out by Reviewer UC3H and include the results in Appendix E.
6. As suggested by T7pn, we added more generation examples of Re-Imagen in Appendix F.
7. As suggested by UC3H, we added imaginary scenes generated by our model in Appendix G.

---

### Author Response · Authors · 2022-11-17
**Connection to Few-Shot In-Context-Learning**

As the discussion deadline is coming. I would like reviewers to check out our newer version of the paper. It includes more analysis and visualization to ablate different factors of our model.

We also compare with DreamBooth in the Appendix. The biggest difference between us and DreamBooth is that we don't need to fine-tune the model on provided image, Re-Imagen can directly ground on the inputs to generate new images. It's very similar to 'prompting'.

Therefore, we also want to draw connections to the few-shot in-context-learning paradigm.

- In the standard in-context learning, the language model will encode a few demonstrations $(x_1, x_2, x_3)$ with encoder $f$, and then the language model cross-attend to the encoded features of $f(x_1, x_2, x_3)$ to generate a completion (text snippet).

- Our model contains a downstack encoder $f(c_p, x_t)$ where $c_p, x_t$ are the text-image pairs and a diffusion-based decoder $g(features)$. By demonstrating a few retrievals/self-provided image-text pairs $(c_p^1, x_t^1), (c_p^2, x_t^2), (c_p^3, x_t^3)$, we apply the encoder $f$ to encode these demonstrations into a sequence of features. The diffusion-based decoder model $g$ will attend to these features to generate a completion (new image).

In essence, Re-Imagen is a type of in-context-learning for image generation.

---

### Decision · Program_Chairs · 2023-01-20

**Decision:**

Accept: poster

**Justification For Why Not Higher Score:**

Some novelty concerns remain.

**Justification For Why Not Lower Score:**

Results still hold value to community.

**Metareview: Summary, Strengths And Weaknesses:**

Paper Summary:
Authors present a modification over Imagen that provides additional context to the diffusion model in the form of images and text from a text-based search of a database. Approach yields SOTA on existing benchmarks, and authors propose a new benchmark that is stratified by frequency in the dataset.


Review Summary:

Pros:

- Simple but effective approach (T7pn, UC3H, oSTW)
- Well written (T7pn, xjWB)
- Popular topic (T7pn, xjWB)
- Well documented limitations (T7pn)
- Good results (xjWB, UC3H, oSTW)

Cons:
- Some descriptions of model architecture not clear (T7pn) -- Authors provided more details.
- Technical contribution / novelty is somewhat limited (T7pn, oSTW) -- Authors admit that while the idea is straightforward, the engineering required to get it to work well was not trivial.
- Needs more visual examples (T7pn) -- Authors added examples.
- Authors are changing the nature of the input by introducing a database, input is now text and database (xjWB, oSTW) -- Authors argue the inference speed is similar and the database is fixed and abstracted away from the user.
- Comparisons could be improved to be more direct/fair (xjWB) -- Authors added smaller model to the comparisons.
- Need more details on design choices (UC3H) -- Authors have responded with more information.

AC Recommendation: Accept. All reviewers except one lean accept because the paper is well written, is simple and effective, and posts good results and discussions. One lean reject reviewer felt the novelty was limited. However AC feels while technical complexity is low, this approach of creating a hybrid between a model and a database retrieval system has not been considered previously and it appears to be a novel perspective on this highly popular topic.


**Note From Pc:**

if the above contains the word "oral" or "spotlight" please see: "oral" presentation means -> notable-top-5% and "spotlight" means -> notable-top-25%. As stated in our emails, we are disassociating presentation type from AC recommendations